# Whole-genome screens reveal regulators of differentiation state and context-dependent migration in human neutrophils

Nathan M. Belliveau[1], Matthew J. Footer [1], Emel Akdoğan [2], Aaron P. van Loon[1], Sean R. Collins [2] & Julie A. Theriot [1] ✉

Neutrophils are the most abundant leukocyte in humans and provide a critical early line of defense as part of our innate immune system. We perform a comprehensive, genome-wide assessment of the molecular factors critical to proliferation, differentiation, and cell migration in a neutrophil-like cell line. Through the development of multiple migration screen strategies, we specifically probe directed (chemotaxis), undirected (chemokinesis), and 3D amoeboid cell migration in these fast-moving cells. We identify a role for mTORC1 signaling in cell differentiation, which influences neutrophil abundance, survival, and migratory behavior. Across our individual migration screens, we identify genes involved in adhesion-dependent and adhesion-independent cell migration, protein trafficking, and regulation of the acto-myosin cytoskeleton. This genome-wide screening strategy, therefore, provides an invaluable approach to the study of neutrophils and provides a resource that will inform future studies of cell migration in these and other rapidly migrating cells.

Among the cells of our immune system, neutrophils are the most abundant cell type and provide a vital early response in host defense by migrating to sites of infection or tissue wounding[1,2]. Paramount to their success is an exquisite sensitivity to chemical gradients, extremely rapid migration speeds on the order of 5–20 μm/min, and an ability to perform directed migration over long distances and through a wide variety of distinct tissue environments[3–5]. Work in recent years has begun to reveal neutrophils as a more heterogeneous cell type than previously thought[2,6,7], though the mechanisms that support differentiation and phenotypic diversity remain incompletely understood. Furthermore, relatively little is known about how specific molecular players may change or adapt as the context and environment of cell migration change.

The emergence of CRISPR-based gene perturbation approaches and robust genome-wide targeted guide libraries now make it possible to perform unbiased functional genomic screens in human cells[8–10]. These approaches offer significant technical improvements over past strategies such as RNAi and offer the opportunity to more comprehensively identify the genes involved in a biological process[11]. However, the use of genome-wide CRISPR-based screens to study complex and dynamic cellular processes has been more limited, with only a few notable exceptions where complex enrichment methods have been applied to identify factors important for phagocytosis and for cell motility[12–15]. The development of new functional screening strategies is expected to provide new biological insights.

Our current understanding of cell migration has relied heavily on videomicroscopy to assess behavior, but that is generally limited to the study of tens or hundreds of individual cells[16,17]. To extend screening tools to perform a comprehensive screen of neutrophil cell migration, there are several notable challenges. First, genome-wide screens require millions of cells, demanding that the assays to assess the relevant biological response be relatively simple and easily scalable[18]. For example, current pooled CRISPR strategies commonly use simple selections such as survival after a drug treatment, or

[1]Department of Biology and Howard Hughes Medical Institute, University of Washington, Seattle, WA 98195, USA. [2]Department of Microbiology and Molecular Genetics, University of California, Davis, Davis, CA 95616, USA. ✉e-mail: jtheriot@uw.edu

enrichments of a cell population with fluorescence-activated cell sorting, to identify genetic perturbations of interest. Second, the terminal differentiation status of neutrophils and their short lifetime, on the order of days[19], limit the use of primary human cells and the possible time scale of individual migration experiments.

In this work, we present the results of several pooled genome-wide CRISPRi screens that provide a comprehensive, genome-wide look at the molecular factors contributing to proliferation, neutrophil differentiation and cell migration. Proliferation and differentiation were assessed by performing pooled dropout assays[18]. Separate migration screens were developed to assess directed migration (chemotaxis), undirected migration (chemokinesis), and 3D amoeboid migration through an extracellular matrix. We confirm known molecular mechanisms contributing to cell proliferation and differentiation and identify an unexpected role for mTORC1 signaling that alters differentiation, survival, and cell migration. We also find a near-perfect correlation between the genes important for chemotaxis and chemokinesis, suggesting that both modes of migration are mechanistically identical. Lastly, we use the results from our different screens of cell migration to distinguish between adhesion-dependent and adhesion-independent cell migration, ultimately identifying several hundred genes that are important across these different migratory contexts. This work demonstrates an invaluable strategy to study cell migration and provides a resource that will apply to future studies of migration in neutrophils and other rapidly migrating cell types.

## Results

### Pooled CRISPRi screens identify genes that alter cell proliferation, neutrophil differentiation, and cell migration

To identify novel regulators important for neutrophil biology, as well as to facilitate our primary goal of identifying genetic factors critical for cell migration, we used the immortalized HL-60 human tissue culture cell line, derived from a patient with acute promyelocytic leukemia[19–21]. The proliferating, undifferentiated HL-60 cells (uHL-60) can be induced to differentiate into a neutrophil-like cell type (dHL-60) by treatment with the signaling molecule all-trans retinoic acid (ATRA) or with the organic solvent dimethylsufoxide (DMSO)[22]. After differentiation, the chemotactic and migratory behaviors of dHL-60 cells closely mimic those of primary neutrophils, and they are able to clear fungal infections in neutropenic mice[23–25]. We validated the efficiency of knockdown in uHL-60 cells expressing dCas9-KRAB by targeting the CD4 gene, with immunofluorescence flow cytometry measurements demonstrating robust knockdown using this construct (Fig. 1a). This dCas9-KRAB construct, which includes a minimal-ubiquitous chromatin opening region and proteolysis-resistant 80 amino acid XTEN linker[26,27], provided substantially better efficacy over other constructs tested in these cells (Supplementary Fig. 1).

We used a pooled genome-wide CRISPRi library (3 sgRNA per gene[10]) to perform dropout-type assays of proliferation and differentiation (Fig. 1b). For proliferation, we quantified changes in sgRNA abundance in uHL-60 cells following six days of growth, as compared to day zero. We identified 2,127 genes that disrupted growth and only 56 genes that enhanced growth (Supplementary Data 1). Our results were well-correlated with those reported by Sanson et al. in HT29 and A375 cell lines using this CRISPRi library (Supplementary Fig. 2a). To derive dHL-60 cells, we induced differentiation by incubating uHL-60 cells with 1.57% DMSO for five days, which provides a near-complete differentiation of the cell population into CD11b+ neutrophil-like cells[28]. Viable dHL-60 cells were isolated using density gradient centrifugation to remove dead cells and cellular debris following our differentiation protocol. We compared sgRNA abundance between dHL-60 and uHL-60 cells to identify gene perturbations that altered the abundance of cells during differentiation and, therefore, could serve as indicators of altered differentiation. Here, we identified 989 genes that were depleted and 869 genes that were enriched relative to our control

sgRNAs (Supplementary Data 2). The ratio of enriched sgRNAs to depleted sgRNAs was strikingly different from our proliferation screen, where only ~2% of knockdowns led to an enrichment of sgRNAs.

We assessed cell migration using three different experimental paradigms. For the first two paradigms, chemotaxis and chemokinesis, we performed scaled-up transwell migration assays (see "Methods"), which mimic migration through tight cellular junctions during transmigration across an endothelial layer[16] (Fig. 1c, left panel). Here, cells were added to the top reservoir above a track-etch membrane with 3 μm diameter pores. To assay these two modes of cell migration, we manipulated the distribution of heat-inactivated fetal bovine serum (hiFBS), a general stimulant for migration[29]. Chemotaxis, or directed migration, was assayed by including 10% hiFBS only in the bottom reservoir, resulting in a chemoattractant gradient toward the bottom reservoir. We separately assayed chemokinesis[30,31], referring to stimulated migration absent of any directional cue, by providing a uniform 10% hiFBS environment in both reservoirs. In both the chemotaxis and chemokinesis assays, cells were collected following periods of two and six hours (Fig. 1d). To assess migratory success, we separately collected both the migratory cells that made it to the bottom reservoir and the cells that remained above the track-etch membrane. To identify significant gene perturbations, normalized $\log_2$ fold-change values were calculated by comparing sgRNA abundances in these cell pools relative to a reference pool of dHL-60 cells (see "Methods" for further details).

Our final migration assay focused on probing amoeboid three-dimensional (3D) migration by embedding cells in a synthetic extracellular matrix (ECM), more representative of migration through the intercellular spaces in tissue[32]. Cells were embedded at the bottom of a thin layer (~200 μm) of collagen ECM that they would need to traverse to reach a second layer of fibrin ECM where they could be recovered (Fig. 1c, right panel). To a first approximation, we expect cells to perform a random walk, whose mean squared displacement will scale linearly with time[33]. Migration through this complex environment will therefore require substantially more time compared to migration through the thin track-etch membranes and we therefore only considered a longer, nine-hour period prior to cell collection in these experiments (Fig. 1d). The most migratory dHL-60 cells were collected by degrading the upper fibrin layer using the enzyme nattokinase, which has protease activity specific to fibrin[34]. We also collected the cells still in collagen, and calculated normalized $\log_2$ fold-change by comparing sgRNA abundances in these cell subpopulations relative to a reference population.

Across our entire set of cell migration assays, we identified 344 genes that reduced the fraction of migratory cells and 31 genes that increased this fraction, relative to migration of control sgRNAs (Fig. 1e). The results of the pooled CRISPRi screens therefore revealed a comprehensive set of genes that play crucial roles in cell proliferation, neutrophil differentiation, and cell migration. We found that nearly half of the genes identified in each screen of proliferation and differentiation were unique to those screens, along with a substantial fraction of genes that affect both of these processes (Fig. 1f). Importantly, a significant number of genes were identified as unique regulators in either differentiation or cell migration, or both. In the sections that follow we characterize the phenotypic changes observed following genetic perturbation across these subsets of genes.

### CRISPRi screens identify genes important for differentiation into migratory neutrophils

Stimulating the differentiation of leukemia cells using pharmacological agents remains a key strategy for the clinical treatment of acute promyelocytic leukemia[35]. We were therefore interested in identifying what gene perturbations influenced the differentiation of uHL-60 cells into migratory, terminally differentiated (non-proliferating) dHL-60 cells. Thus, we began by performing gene set enrichment analysis (GSEA)[36] to identify the pathways and gene categories that were overrepresented

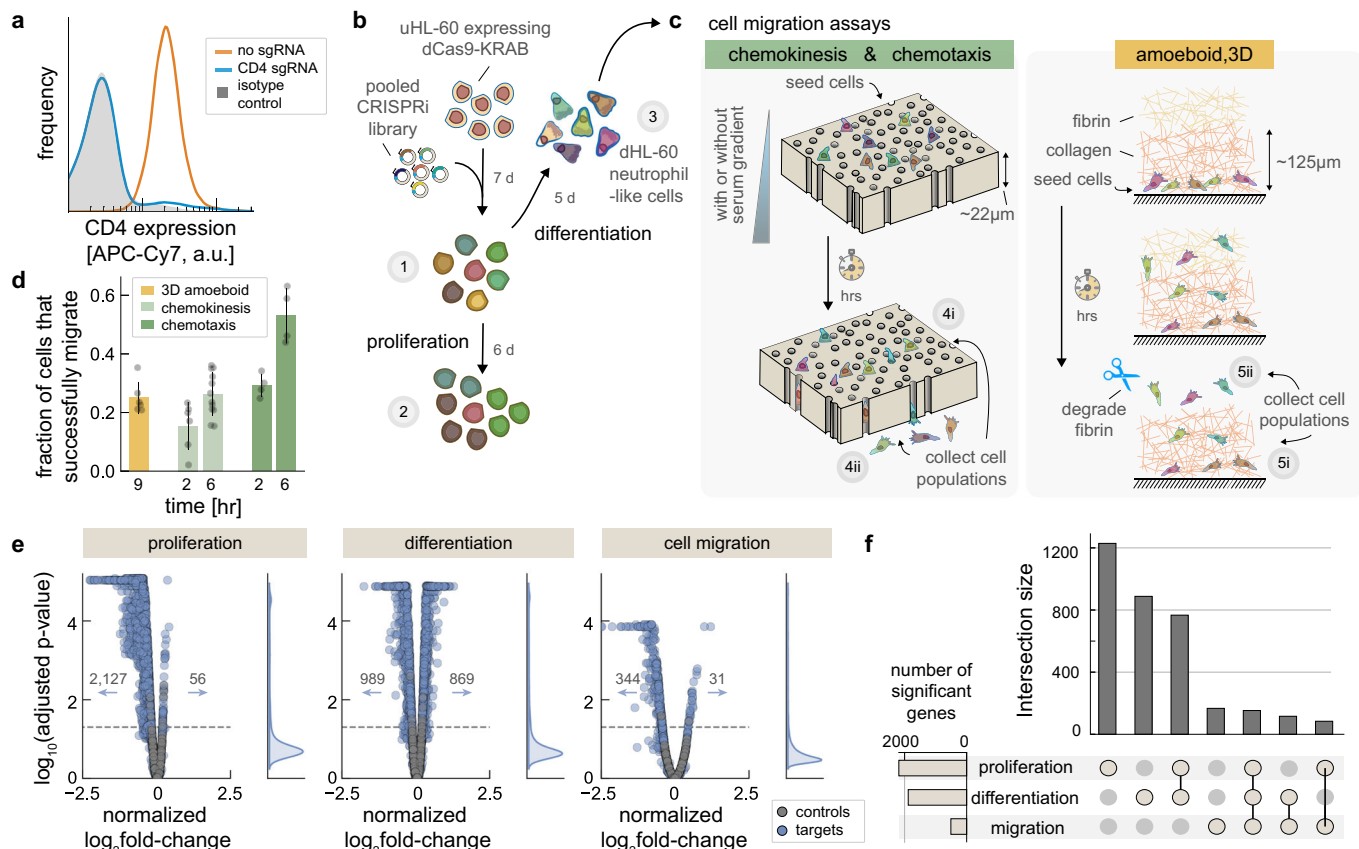

**Fig. 1 | Genome-wide CRISPRi screens of proliferation, differentiation, and cell migration. a** Flow cytometry immunofluorescence shows near-complete loss of CD4 protein (blue) in uHL-60 cells expressing dCas9-KRAB, relative to normal expression (orange) and an isotype control (gray, shaded). **b** Schematic of pooled genome-wide CRISPRi dropout experiments of uHL-60 cell proliferation and differentiation into dHL-60 neutrophils. Proliferation was assayed by comparing sgRNA abundances following six days of growth (-24 hr doubling time) (set 2 versus set 1; 4 independent replicates). Differentiation was assayed by comparing sgRNA abundance between dHL-60 neutrophils and uHL-60 cells (set 3 versus set 1; 8 independent replicates). **c** Schematic of pooled CRISPRi cell migration assays. Migration of dHL-60 cells were assayed across three experiments: chemotaxis (serum gradient), chemokinesis (uniform serum stimulation), and 3D amoeboid migration in an extracellular matrix (see "Methods"). For quantification, sgRNA abundance in both migratory fractions (sets 4i and 5i) and remaining cells (sets 4ii and 5ii) were compared to our initial dHL-60 library (set 3). Membrane, pores and cells drawn to scale. **d** Error bars represent mean values +/− SD of the migratory

fraction across independent experiments (3D amoeboid: 6 replicates; chemokinesis 2 hr and 6 hr: 4 replicates each; chemotaxis 2 hr: 4 replicates; chemotaxis 6 hr: 16 replicates). For 3D amoeboid experiments, the migratory fraction of cells was collected from the fibrin layer. **e** Volcano plots across the screens of proliferation, differentiation, and cell migration. Data points represent the average log2 fold-change from three sgRNAs per gene across independent experiments (4 replicate screens for proliferation, 8 replicate screens for differentiation, and 20 migration screens). Cell migration values represent an average across all migration assays. Controls were generated by randomly selecting groups of three control sgRNAs. P-values were calculated using a one-sided permutation test, adjusted for multiple comparisons using the Benjamini–Hochberg procedure (dashed line: $p = 0.05$). **f** Screen overlap. The number of significant genes are identified in the left horizontal bar plot using an adjusted $p$ value cutoff of 0.05, while the intersection of genes across screens is shown in the vertical bar plot (dot diagram identifies the specific intersection).

---

among the differentiation screen data (Fig. 2a). We found a positive enrichment across various metabolic processes, particularly genes involved in oxidative phosphorylation (electron transport chain, mitochondrial protein synthesis). We also found a depletion of key regulatory genes associated with granulopoiesis including the transcription factors Fli-1 (FLI1) and the CCAAT/enhancer binding proteins, C/EBPα (CEBPA) and C/EBPε (CEBPE), suggesting that our screen data is identifying genes specific to neutrophil differentiation.

To further distinguish genes whose knockdown specifically affected differentiation per se from those that generally perturbed basic cellular processes, we also plotted the differentiation screen's log2 fold-change values against both our screens of proliferation and the migratory data, averaged across all migration screens (Fig. 2b). PU.1 (SPI1), another transcription factor that is highly expressed in neutrophils[37,38], was identified across all screens and was among the strongest perturbations to migration (normalized log2 fold-change of −1.2, Fig. 2b right panel). Strikingly, knockdown of genes

associated with oxidative phosphorylation and mitochondrial translation showed systematic effects in all three screens, with sgRNAs associated with this process mostly enriched following differentiation, while mostly depleted in the screens for both proliferation and migration (Fig. 2b, red and green points). In contrast, sgRNAs targeting genes associated with mammalian target of rapamycin (mTOR) signaling were enriched following cell differentiation and depleted in the migration assays, but knockdown of these genes had no consequence on proliferation (Fig. 2b, yellow points).

One unexpected gene identified in both our differentiation and migration screens was ATIC (Fig. 2b), which codes for an enzyme that acts on the adenosine monophosphate analog AICAR, an intermediate in the generation of inosine monophosphate in the purine biosynthesis pathway[39]. We constructed a stable cell line expressing a sgRNA targeting ATIC and found a substantial number of polarized migratory cells prior to induction of differentiation, a notable phenotype that is not observed in unperturbed uHL-60 cells (Fig. 2c and Supplementary

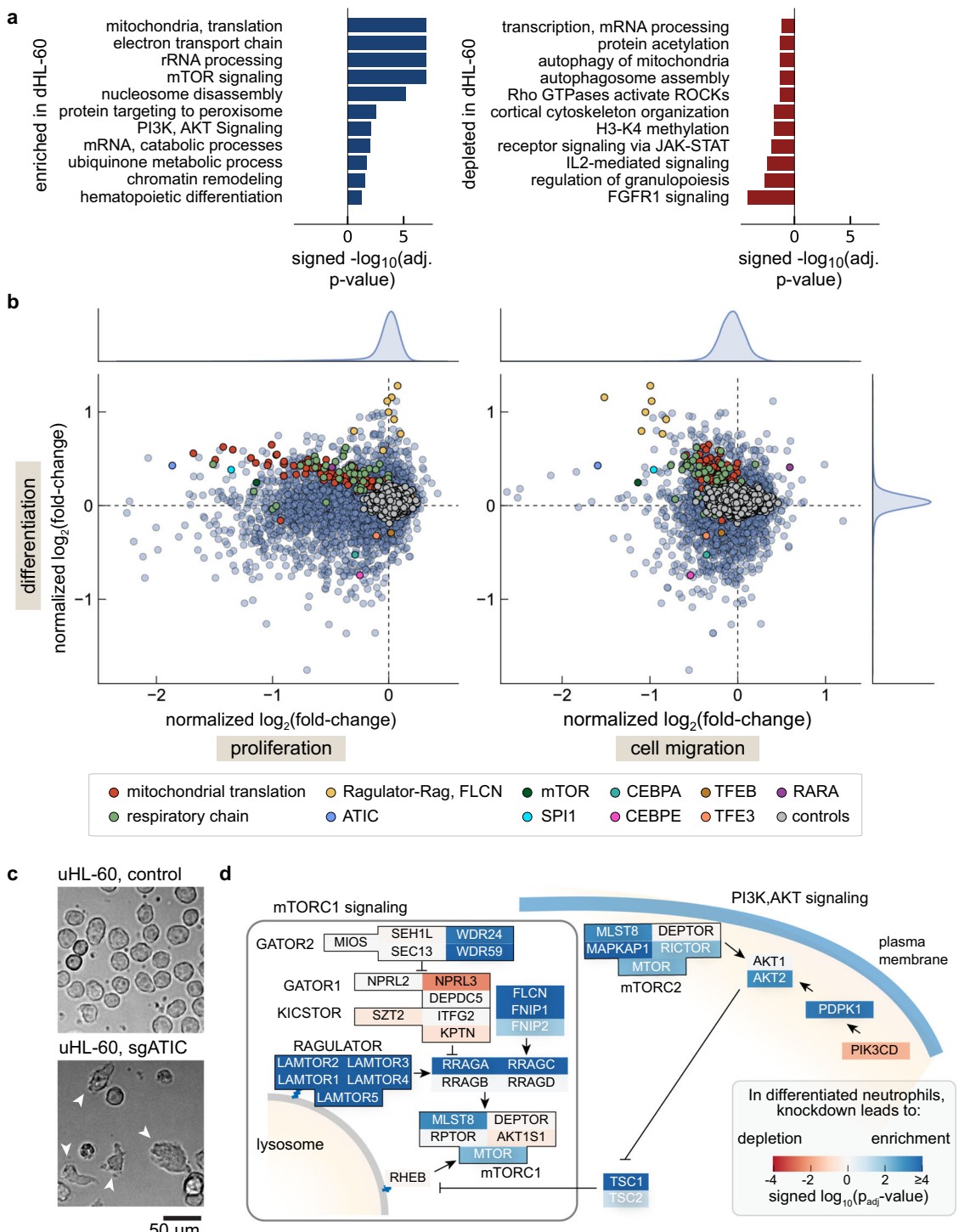

**Fig. 2 | Identification of genes and pathways important for neutrophil differentiation.** **a** Pathways enriched in our CRISPRi differentiation screen (dHL-60 cells relative to uHL-60 cells). Pathways that were associated with genes whose knockdown predominantly led to an enrichment of target sgRNAs in the dHL-60 cell population are identified in blue, while those that decreased in abundance are in red. P-values estimate the statistical significance of gene set enrichment, calculated using a one-sided permutation test and adjusted for multiple comparisons using the Benjamini–Hochberg procedure. **b** Comparison of log$_2$ fold-changes across the CRISPRi screens of proliferation, differentiation, and cell migration. Several gene sets identified through our pathway enrichment analysis and several known regulators of neutrophil differentiation are identified. The cell migration

data points represent normalized log$_2$ fold-change values, calculated by averaging across all individual migration screen replicates. **c** Brightfield microscopy of uHL-60 CRISPRi knockdown lines targeting ATIC and a control sgRNA. Control uHL-60 cells exhibit an expected round morphology, while sgRNA targeting of ATIC resulted in many cells that exhibited a migratory capability (white arrows). Images are representative of acquisitions across three fields of view. **d** Schematic of mTORC1/mTORC2 signaling pathway, color coded by signed statistical significance values (log$_{10}$ $p_{adj}$ value) from the differentiation screen results. Blue indicates gene targets whose sgRNA were enriched in the dHL-60 cells, while red indicates those that were depleted.

Fig. 2b). AICAR is capable of stimulating AMP-dependent protein kinase (AMPK) activity[40,41] and it is possible that ATIC knockdown changes the basal concentration of AICAR, which may alter the cell's metabolic and energy state in a way that drives differentiation. Consistent with these observations, RNA transcriptome analysis showed that these cells had a transcriptional profile more similar to that of dHL-60 cells expressing a control sgRNA than to control uHL-60 cells (Supplementary Fig. 2d). More work will be needed to understand the broader impact of this gene perturbation on cell migration.

Considering our differentiation screen results more broadly, we found that many of the identified genes were also enriched in recent genome-scale efforts to characterize mouse embryonic stem cell differentiation[42,43]. Notably, of the roughly 500 genes reported as important for exit from pluripotency in mouse embryonic stem cells, half were present in our differentiation data set (adjusted p value <0.01 when applying gene set enrichment analysis, Supplementary Fig. 2b). While embryonic stem cells and uHL-60 cells represent distinct developmental cell stages, this commonality suggests shared processes that may be important as mammalian cells change their proliferative status and undergo state transitions in differentiation.

## Disruption of folliculin and Ragulator-Rag signaling pathways potentiate survival of dHL-60 neutrophils

Given the enrichment of sgRNAs associated with mTOR signaling during both differentiation and migration, we wanted to further understand the consequence of these gene perturbations. In humans, the protein kinase mTOR is a component of two distinct complexes, mTORC1, and mTORC2. While mTORC2 has previously been implicated in cell migration and chemotaxis[44–48], we were surprised to find many genes associated with mTORC1 signaling in our screens of differentiation and migration (Fig. 2d). mTORC1 coordinates cell growth through its activity on the surface of lysosomes by mediating cellular changes in translation regulation, metabolism, and autophagy[49]. The genes most enriched in the differentiation screen were directly upstream of mTORC1 and included the Rag guanosine triphosphatase (GTPase) A/B:C/D heterodimer, which recruits mTORC1 to the lysosomal membrane via binding to the Ragulator complex (LAMTOR1-5)[50]. Knockdown of folliculin (FLCN), a GTPase-activating protein that targets RagC/D and promotes an active state of the Rag heterodimer[51], led to a similar enrichment of dHL-60 cells in the differentiation screen and depletion in the migration screen.

To characterize how these gene knockdowns might affect differentiation, we generated individual stable cell lines expressing mTORC1-related sgRNAs targeting LAMTOR1, FLCN, TSC1, and also a cell line expressing an sgRNA targeting RICTOR, a key subunit of the mTORC2 complex[47]. Following the initiation of neutrophil differentiation, all knockdown cell lines showed a higher cell density compared to a control sgRNA cell line after comparable time periods, consistent with their overrepresentation in our differentiation screen results (Fig. 3a). Normally, cell densities stopped increasing by about 4 days after initial DMSO-induced differentiation and began to decline, presumably due to apoptosis of the terminally differentiated dHL-60s (Fig. 3b). Interestingly, LAMTOR1 and FLCN knockdown lines showed a distinct increase in cell lifetime (Fig. 2e, pink and purple symbols). To further confirm a functional role for mTORC1, we treated these two knockdown lines with 10 nM and 100 nM rapamycin, dosages expected to abolish mTORC1 kinase activity[52]. Rapamycin treatment resulted in a decrease in the survival of these cells following differentiation, restoring their survival characteristics to the lower levels associated with the sgRNA control cell line (Fig. 3c).

We further assessed the role of mTORC1 on differentiation by quantifying key molecular markers of neutrophils. Here we used flow cytometry to measure the induced surface expression of CD11b, an early differentiation marker also known as integrin $\alpha_M$ (ITGAM), and the fMLF receptor (FPR1) that recognizes chemoattractant

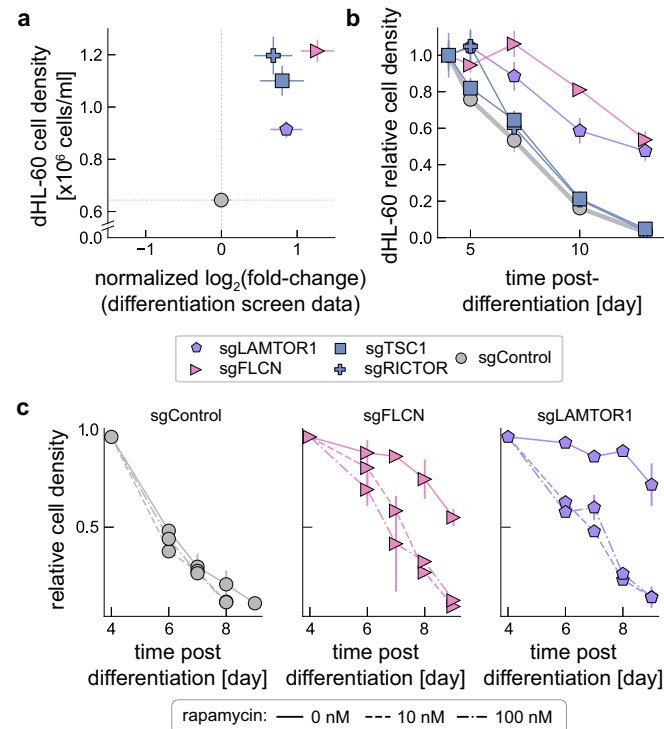

**Fig. 3 | Characterization of cellular growth and survival following knockdown of mTOR-related genes. a** Differentiation screen results were confirmed across a number of sgRNA targets. Cell density was monitored at 5-days following the initiation of neutrophil differentiation. The dashed lines represent the values obtained for dHL-60 cells with a control sgRNA. Error bars represent mean values +/− SD across independent experiments (8 experiments for screen and 3 experiments for cell density measurements). **b** Change in dHL-60 cell density following differentiation. Cell density measurements were normalized to day 4, following initiation of cell differentiation of uHL-60 cells. Error bars represent mean values +/− SD across 3 independent experiments. Note that media was replenished every three days, with cell density measurements corrected to account for changes in media volume and evaporation. **c** Rapamycin treatment in dHL-60 CRISPRi knockdown lines targeting FLCN, LAMTOR1, and a control sgRNA. Treatment of dHL-60 cells was begun on day 4 post-initiation of differentiation. Cells were either untreated (solid line), or treated with rapamycin at 10 nM rapamycin (dashed lines) and 100 nM rapamycin (dash-dot lines). Error bars represent mean values +/− SD across 3 independent experiments.

N-formylated peptides[53]. Both markers showed little to no expression in uHL-60 cells but were strongly induced in our dHL-60 cells (Fig. 4a). As positive controls for disruption of neutrophil differentiation, we constructed stable cell lines expressing sgRNAs targeting the two essential differentiation genes, SPI1 and CEBPE. To assess induction of differentiation markers, we used principal component analysis to quantify the axis associated with co-induction of the two surface markers ITGAM and FPR1 (principal mode 1, Fig. 4b). As expected, sgRNAs targeting SPI1 and CEBPE showed reduced induction of differentiation markers. In contrast, we found that stable cell lines expressing sgRNAs targeting LAMTOR1 and FLCN exhibited higher induction of the differentiation markers as compared to controls (Fig. 4c and Supplementary Fig. 3a). These results show that cells expressing these sgRNAs are still undergoing terminal differentiation but with altered survival characteristics.

## Disruption of folliculin and Ragulator-Rag signaling pathways results in altered but active mTORC1 signaling in dHL-60 neutrophils

In order to identify the specific changes in mTOR signaling associated with knockdown of LAMTOR1 and FLCN, we began by checking for

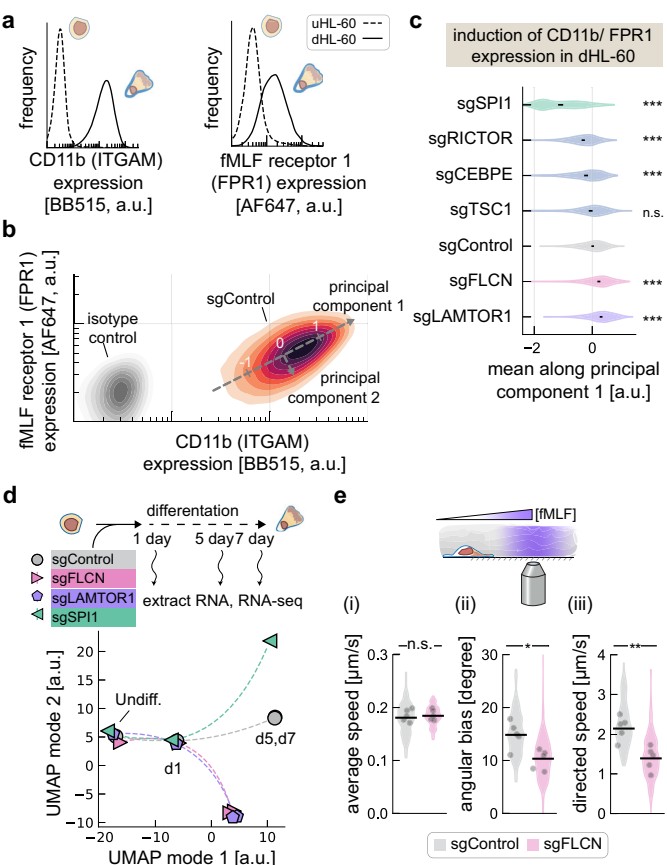

**Fig. 4 | Knockdown of FLCN and LAMTOR1 alters differentiation trajectory and results in cells with poorer chemotactic sensitivity. a** Flow cytometry immunofluorescence measurements of CD11b (ITGAM) and fMLP receptor 1 (FPR1) cell surface expression in uHL-60 and dHL-60 cells. **b** The two-dimensional heatmap shows the induced expression of CD11b and fMLP receptor 1 in dHL-60 cells. The axis associated with induction of these surface markers were identified by applying principal components analysis. Measurements using isotype control antibodies are shown in gray. **c** The first principal component identified in (**b**) was used to compare changes in expression induction in different gene knockdown lines. Black lines indicate the 99% confidence interval for the log expression mean along the first principle component, calculated by bootstrapping across single-cell flow cytometry measurements. A two-sided Mann-Whitney U test was applied to the bootstrapped log expression values from each knockdown cell line and the control cell line (***$p < 0.001$). **d** Transcriptional changes following knockdown of FLCN, LAMTOR1, and SPI1 were assayed by RNA-seq pre-differentiation (undiff.), 1-day, 5-day, and 7-day post-differentiation. Dimensionality reduction using UMAP was applied to transcription data and pseudo-plotted using a spline to show temporal trajectory. Individual data points represent an average across 6 independent RNA-seq samples. **e** The acute chemotaxis response of dHL-60 cells was assayed by photo-uncaging fMLP during migration of agarose-confined cells on BSA passivated coverslips. Average instantaneous speed (i), angular bias (ii), and the directed speed (projected speed along direction of fMLP gradient) (iii) are shown. Data points indicate mean across 5 independent experiments (~3500 cells per cell line, per experiment), with the shaded regions showing the distribution across measurements. A two-sided Mann–Whitney $U$-test indicated a significant difference in angular bias (*$p = 0.03$) and directed speed (**$p = 0.008$).

altered phosphorylation of known targets of mTORC1 and mTORC2 protein kinase activity. Western blot analysis of ribosomal S6 kinase and Akt, which are well-characterized targets of mTORC1 and mTORC2, respectively, showed that kinase activity of mTORC1 and mTORC2 remained active following knockdown of LAMTOR1 and FLCN (Supplementary Fig. 3b, c). It is also known that mTORC1 can regulate the activity of transcription factors TFEB and TFE3, which depend more specifically on the activity of RagC/D[49,54]. Notably,

knockdown of TFEB and TFE3 resulted in a modest but significant decrease in sgRNA abundance in our differentiation screen (Fig. 2b). We, therefore, turned to whole-transcriptome sequencing (RNA-seq) to identify global transcriptional changes that might provide more insight into the changes in mTORC signaling associated with knockdown of LAMTOR1 and FLCN.

Focusing on the stable cell lines expressing sgRNAs targeting LAMTOR1, FLCN, and SPI1, we performed RNA-seq on uHL-60 cells, and dHL-60 cells at days 1, 5, and 7 following the initiation of differentiation. Using dimensionality reduction (UMAP[55]) to take a broad look at the entire data set, we found similar changes in transcription in our LAMTOR1 and FLCN knockdown lines, which were distinct from both the line expressing our control sgRNA and the line expressing SPI1 sgRNA (Fig. 3d). Also consistent with our screen results, the transcriptional profiles across our knockdown lines were fairly similar to the controls in undifferentiated cells and one day after induction of differentiation, only diverging later in the differentiation process, suggesting that these gene perturbations are specifically involved in differentiation and neutrophil function.

To better understand why the FLCN and LAMTOR1 knockdown cells enjoyed a prolonged lifespan, we delved more deeply into the gene expression data. Relative to control cells, we found enrichment for genes associated with lysosomes, autophagy, and transcription of ribosomal genes (Supplementary Fig. 3d, e). Importantly, these match the reported roles of TFEB and TFE3 as master regulators of lysosomal biogenesis and autophagy[49], further confirming that altered mTORC1 signaling is indeed along the RagC/D-FLCN axis. We found a substantial decrease in expression of the autophagy-activating kinases ULK1 and ULK2, and increased expression of the anti-apoptotic gene BCL2 (Supplementary Fig. 3e), which may support the extended survival of these cells. Direct inhibition of autophagy has been shown to affect both neutrophil differentiation and effector function[56,57] and our data suggest that FLCN and LAMTOR1 knockdown play a similar inhibitory role through altered mTORC1 activity. These knockdown lines also showed changes in the expression of genes associated with neutrophil degranulation, including an increase in *mpo* and a decrease in *mmp9* (Supplementary Fig. 3d and Supplementary Data 3 and 4). This is also observed following the inhibition of autophagy during differentiation and may be a reflection of incomplete differentiation[57].

Intriguingly, with respect to differentiation, our characterization of cell surface markers showed that the FLCN and LAMTOR1 knockdown lines expressed normal or slightly elevated levels of fMLF receptor as compared to controls (Fig. 4c and Supplementary Fig. 3a). We were therefore interested in whether these cells maintained their sensitivity to fMLF as a chemotactic agent, and explored this further in our FLCN knockdown line. Here we used photo-activation of caged fMLF to generate spatial gradients of the small chemotactic peptide, using a standard assay where migratory cells are sandwiched between a BSA-coated coverslip and an agarose overlay to minimize requirements for adhesion[30]. While FLCN knockdown cells migrated with similar speeds as our control sgRNA line in this context (Fig. 3h (i)), knockdown cells were much less responsive to fMLF. Across cells, we observed a reduced average angular bias of about 10°, compared to 15° for our control sgRNA line (Fig. 3h (ii)). This also resulted in a reduced directed speed (i.e. projected speed along the spatial fMLF gradient) (Fig. 3h (iii)), showing that FLCN knockdown cells were not as responsive and migratory toward the fMLF gradient.

In our differentiated FLCN and LAMTOR knockdown lines, we also observed an increase in expression of several genes associated with macrophage differentiation, including *ccr5, cd163, cd64, cd71*. uHL-60 cells are multipotent cells that can be differentiated into other cell types, including macrophages[58]. Given the observed induction of CD11b and fMLF receptor in dHL-60 FLCN and LAMTOR1 knockdown lines (Fig. 4c and Supplementary Fig. 3a) and that these cells are longer lived, we reasoned that FLCN and LAMTOR1 knockdown might be

skewing their differentiation trajectory away from a purely neutrophil-like character and towards a more macrophage-like state. To test this possibility, we mined an available RNA-seq dataset that measured gene expression changes during differentiation of uHL-60 cells into both neutrophil-like cells and macrophage-like cells[38], we identified a variety of genes including *cd52* that show higher differential expression in macrophage-like cells (Supplementary Table 1). Using flow cytometry, we found that both FLCN and LAMTOR1 knockdown cell lines exhibited higher expression of CD52 than control lines after DMSO-triggered differentiation, consistent with this hypothesis (Supplementary Fig. 4). While further work is needed to fully dissect the possibility that FLCN and LAMTOR1 knockdown alters the trajectory of cell fate, it does help to explain the observed pattern of enrichment of dHL-60 cells in our differentiation screen.

Overall, we found that disruption of mTORC1-related genes alters (but does not eliminate) mTORC1 signaling. This results in an apparent inhibition of autophagy and reduction in apoptotic signals that extend cell survival, increasing their abundance within the population, but also perturbs their differentiation into fully chemotactic neutrophils.

## Migration through track-etch membranes is dominated by genes associated with cell adhesion

To identify genes important for cell migration through the narrow pores of track-etch membranes, we compared the results of our screens for chemokinesis (with serum present in both top and bottom reservoirs) and chemotaxis (with serum present in the bottom reservoir only). The gene perturbations most detrimental to migration in both our chemotaxis and chemokinesis screens were associated with inside-outside $\alpha_M\beta_2$ integrin signaling (Fig. 5a). Integrins facilitate cell-substrate binding and indeed, when examined migration on a fibronectin-coated coverslip of a stable knockdown line expressing a sgRNA targeting ITGB2, cells exhibited a polarized morphology but were only loosely adherent (Fig. 5b, Supplementary Fig. 4a, and Supplementary Movie 1). We also directly measured the adhesion phenotype by performing an adhesion assay where cells were allowed to adhere to the surface of a plastic culture dish. We found a substantial reduction in the fraction of adherent cells in our ITGB2 knockdown line relative to our control cell line (Supplementary Fig. 5c).

Interestingly, earlier we noted that our LAMTOR1 and FLCN knockdown lines had slightly elevated levels of CD11b (integrin $\alpha_M$) (Fig. 4c), suggesting that these cells may also exhibit an altered adhesion phenotype. Indeed, when placed on a fibronectin-coated coverslip, these cells appeared to make more extensive contact with the substrate and often lacked a normal front-back polarity (Fig. 5c and Supplementary Movie 2). We performed tracking of cell nuclei as cells migrated on fibronectin-coated coverslips for 30 min. Here we found that the LAMTOR1 and FLCN knockdown cells moved at about half the speed of our control sgRNA line, with an average speed of 0.13 μm/s as compared to 0.21 μm/s (Fig. 5d, top). This effect was almost as detrimental as directly knocking down ITGB2 (Fig. 5d, top). These observations are consistent with previous results in other cell types indicating that there is an optimum degree of cell-substrate adhesion for efficient migration, with either increasing or decreasing adhesion causing decreased cell speed[59–62].

Efficient directional migration for motile cells depends on directional persistence as well as cell speed. We employed a Bayesian inference algorithm based on a model for a heterogeneous random walk[63] to calculate a migratory persistence metric for each cell. In this model, a persistence value of zero corresponds to a non-persistent, diffusive movement. Persistence values closer to −1 indicate anti-persistent, reversive movement, while values closer to +1 indicate more persistent migration. In line with the reduced front-back polarity, FLCN and LAMTOR1 knockdown lines showed a reduction in persistence (Fig. 5d, bottom) compared to control cells. These results highlight the

importance of adhesion as a key prerequisite to entry and migration through the pores of the track-etch membrane.

## Chemotaxis and chemokinesis migration screens are strongly correlated

One of the most notable observations from the chemotaxis and chemokinesis datasets was a strong correlation between their normalized $\log_2$ fold-change values (Fig. 5e, left panel; $\rho = 0.99$). This suggests there are no distinct molecular pathways that neutrophils require for directed migration as opposed to random migration, in the context of serum stimulation.

In order to confirm the measured $\log_2$ fold-change values for cells reaching the lower reservoir in the large-scale screen with a more direct measurement of transmigration, we assayed the fraction of cells that migrate across the track-etch membrane in individual knockdown cell lines. We generated individual lines expressing one of the sgRNAs, chosen to span the range of observed $\log_2$ fold-change values from the screen (ITGB2, APBB1IP, TLN1, VPS29, ARHGAP30, FMNL1, ATIC, GIT2). Quantifying the fraction of cells that migrated through the track-etch membrane after two hours with 10% hiFBS added to the bottom reservoir, we found a strong correlation with the $\log_2$ fold-change values from our chemotaxis screen (Fig. 5f, $\rho = 0.87$). More specifically, the strongest perturbation, targeting knockdown of ITGB2, led to only 6% of the cells in the bottom reservoir versus 30% of the cells with our control cell line. Knockdown of GIT2 showed the largest positive increase, with 34% of the cells collected in the bottom reservoir.

Since most gene knockdowns decreased the fraction of cells migrating through the track-etch membrane, we were intrigued by the subset of genes that exhibited a positive $\log_2$ fold-change; that is, those whose knockdown enhanced cell migration. Among the most positively enriched genes was GIT2 (Fig. 5e), which encodes a protein that binds to the p21-activated kinase-interacting guanine nucleotide exchange factors α-PIX (ARHGEF6) and β-PIX (ARHGEF7). α-PIX and β-PIX both enhance the activity of the Rho GTPases Cdc42 and Rac1[64,65] that act as master regulators to enhance actin assembly at the leading edge of motile cells[66]. Along with GIT2, knockdown of α-PIX and another α-PIX binding partner, PPM1F, also exhibited positive enrichment in our migration screens (Fig. 5e). To better understand how knockdown of GIT2 influenced cell migration, we directly examined the motility behavior of our stable cell line with a sgRNA targeting GIT2. Analyzing cell tracks as cells migrated on fibronectin-coated coverslips, we find that GIT2 knockdown cells migrated with an average speed of 0.25 μm/s, or about 20% faster than our control cell line (Fig. 5g, left). Migration otherwise appeared similar to control cells, exhibiting similar migratory persistence (Supplementary Fig. 5d).

In sum, the most substantial gene perturbations across our chemotaxis and chemokinesis screens are those that impact cell–substrate adhesion or play a role as regulatory components governing the behavior of the actomyosin cytoskeleton.

## CRISPRi screen identifies genes important for 3D amoeboid cell migration

To explore how 3D amoeboid migration differs from 2D migration in our track-etch membrane assays, we began by comparing our 3D amoeboid screen with the chemokinesis screen results. We find that many of the adhesion-related genes did not exhibit strong phenotypes in the 3D migration assay (Fig. 6a). This observation is consistent with the expectation for integrin-independent migration of cells embedded in fibrous ECM[67]. Interestingly, knockdown of talin 1 (TLN1), which mediates the linkage between integrins and the actin cytoskeleton, still inhibited migration into the upper fibrin layer of the 3D amoeboid screen. This suggests additional roles for talin 1 beyond the interaction between the actin cytoskeleton and $\alpha_M\beta_2$ integrins in dHL-60 cells.

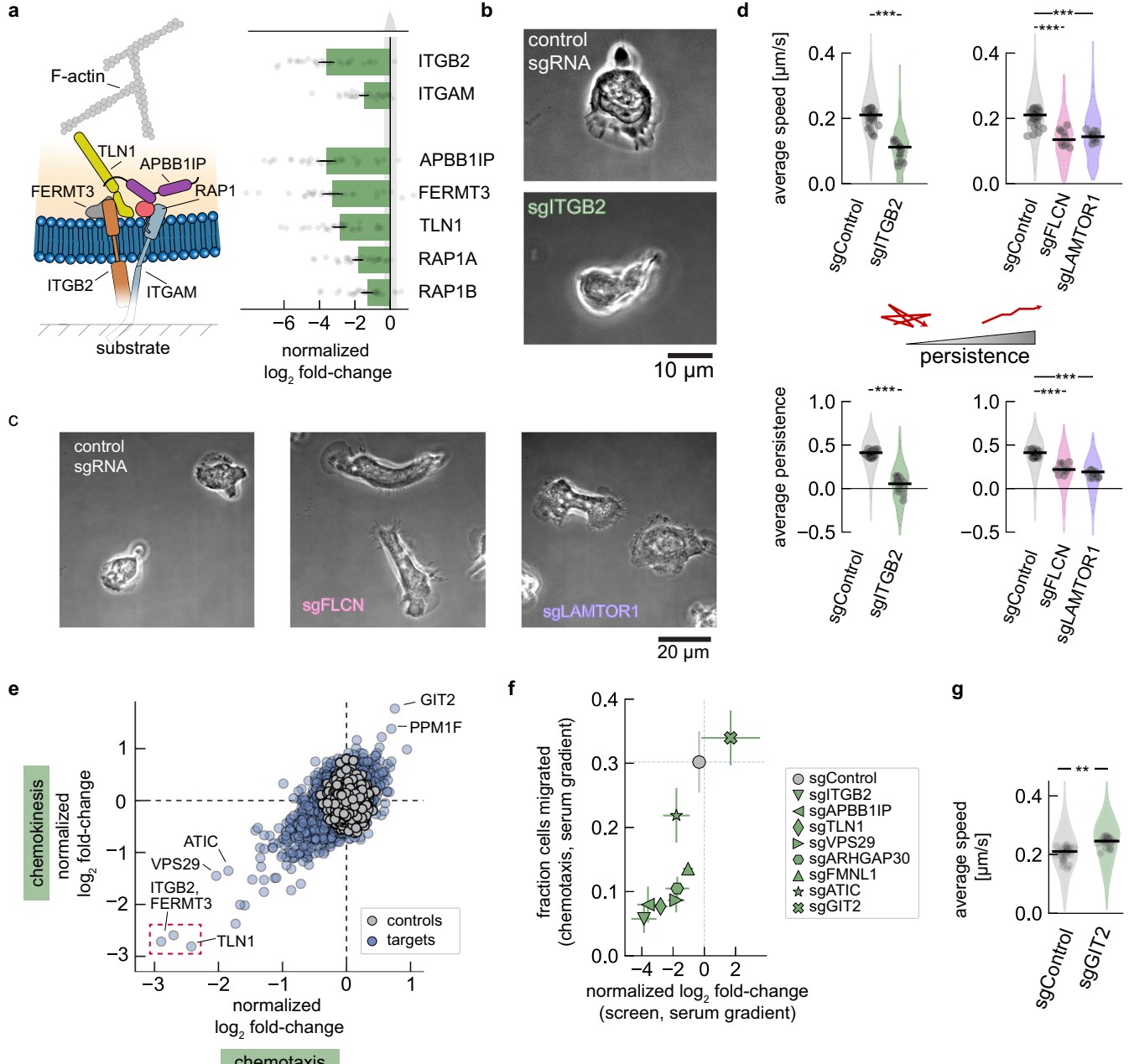

**Fig. 5 | Cell migration CRISPRi screen identifies genes important for adhesion and migration on 2D surfaces. a** Left, components of inside-out $\alpha_M\beta_2$ integrin signaling. Right, normalized log$_2$ fold-changes values for most significant sgRNA in the chemotaxis and chemokinesis screens. Error bars represent mean values +/− SEM across $n = 28$ measurements from 14 independent experiments. The gray shaded region shows the histogram of control sgRNAs. **b** Representative phase images of cell migration on fibronectin-coated coverslips (ITGB2 sgRNA and control cells). Three fields of view were collected for each cell line. **c** Representative phase images of cell migration on fibronectin-coated coverslips (FLCN sgRNA, LAMTOR1 sgRNA, and control cells). Two fields of view were collected for each cell line. **d** Characterization of cell migration phenotypes. Speed was calculated by tracking cell nuclei during migration on fibronectin-coated coverslips. Persistence was inferred from the cell velocity data as described by Metzner et al. (see "Methods"). Measurements represent experiments performed over 2–3 days, acquired across 32 (sgControl), 10 (sgFLCN), and 14 (sgLAMTOR1) fields of view.

Differences were identified using a two-sided Mann–Whitney $U$-test (***$p < 0.001$). **e** Comparison of normalized log$_2$ fold-changes across the pooled CRISPRi cell migration screens of chemotaxis and chemokinesis. ITGB2, FERMT3 and TLN1 genes are identified in red. **f** Comparison between chemotaxis screen normalized log$_2$ fold-changes and measurements of migration fraction of individual knockdown lines exposed to a serum gradient with 10% hiFBS, for 2 h. The gray data point and dashed lines represent the values obtained for control cells. Error bars represent mean values +/− SEM across independent experiments (8 screen replicates and 4 replicates for stable knockdown lines). **g** Characterization of cell migration speed following knockdown of GIT2 from nuclei tracking during migration on fibronectin-coated coverslips. Measurements represent experiments performed over 3 days, acquired across 29 fields of view. The data was compared using a two-sided Mann–Whitney $U$-test (**$p < 0.01$). For (**d**–**g**), individual data points represent average values for cells across a single field of view, with the shaded regions showing the distribution of all measurements.

Of the many interesting candidates, we chose to further characterize two actin regulatory proteins identified in the 3D amoeboid screen, formin-like 1 (FMNL1) and coronin 1A (CORO1A). These proteins have previously been implicated in cell migration[68–70], but their

role during 3D migration remains less well-characterized. We generated stable cell lines with sgRNA targeting each of these genes and quantified migratory speed and persistence as cells migrated in 3D collagen gels. As expected, knockdown of ITGB2 showed no effect on

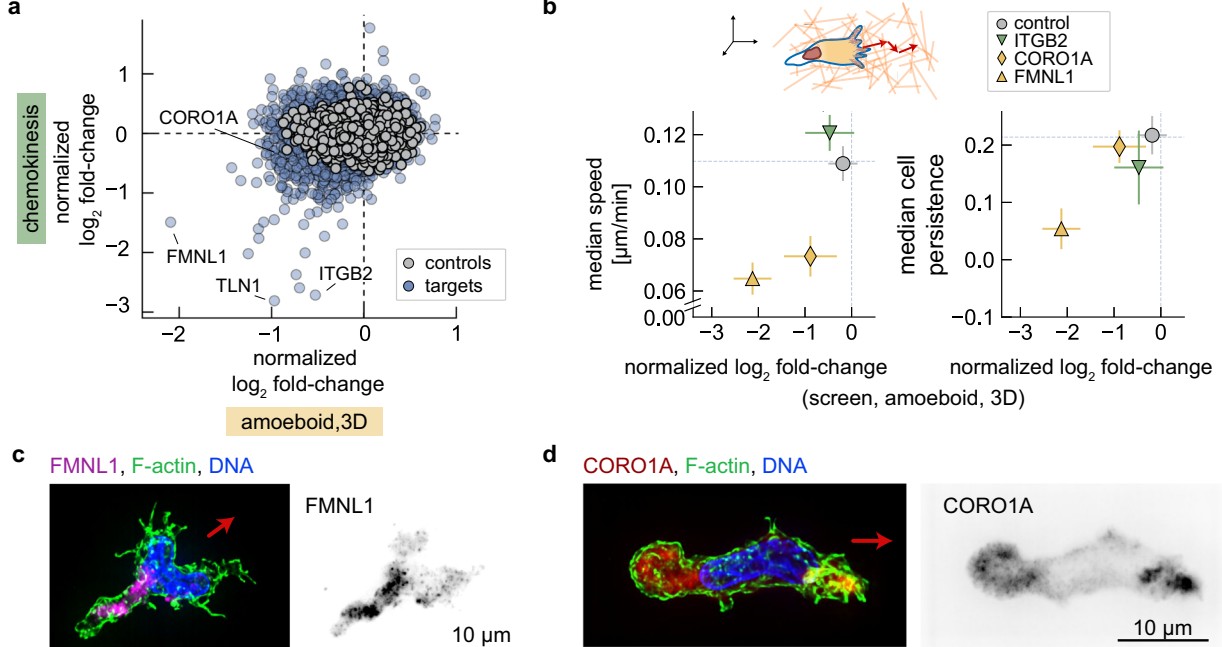

**Fig. 6 | Cell migration CRISPRi screen identifies genes important for 3D amoeboid migration. a** Comparison of normalized log$_2$ fold-changes across the pooled CRISPRi cell migration screens of 3D amoeboid migration and chemokinesis. **b** Comparison of 3D migration normalized log$_2$ fold-changes with measurements of cell speed and migratory persistence via single-cell nuclei tracking. Cells from individual sgRNA knockdown lines were tracked during migration in collagen for 60 min (1 min frame rate). The median cell speed (left) and inferred migratory persistence (right, see "Methods") are plotted against their measured normalized log$_2$ fold-change from the pooled screen. Error bars represent mean values +/− SEM across independent experiments (6 screen replicates and 4 for experiments using

stable knockdown lines). Individual cell line measurements represent experiments performed over 2–5 days, acquired across 10 (sgControl), 7 (sgFLMN1), 7 (sgCORO1A), and 4 (sgITGB2) fields of view. **c**, **d** show immunofluorescence localization of FMNL1 and CORO1A in amoeboid-migrating cells in collagen. F-actin was labeled by phalloidin, while DNA was stained by DAPI. Left images are maximum projection composite images; right images show grayscale localization of formin-like 1 (**c**) and coronin 1A (**d**). Red arrows indicate the approximate direction of cell migration based on cell shape and a more intense phalloidin intensity expected at the cell front.

speed or persistence in 3D, but migration speed was decreased in the FMNL1 and CORO1A knockdown lines, consistent with the results of our 3D screen (Fig. 6b, left panel, $\rho = 0.83$). For both knockdown lines, cells migrated with an average speed of approximately 0.07 µm/s, roughly half as fast as our control sgRNA or ITGB2 knockdown lines, which have average speeds of 0.11–0.12 µm/s.

While knockdown of either FMNL1 or CORO1A resulted in reduced speed, only FMNL1 knockdown showed a significant reduction in migratory persistence (Fig. 5b, right panel). This may reflect different roles during 3D migration. We used immunofluorescence to determine the localization of these proteins in wild-type cells. In contrast to the expected leading edge localization of well-characterized formins like mDia1/2[71], FMNL1 was rear-localized and often directly behind the nucleus (Fig. 6c and Supplementary Fig. 6a). This is consistent with recent work in T cells[72], who found similar localization and hypothesized that formin-like 1 may support actin polymerization to aid in squeezing the nucleus through tight endothelial barriers. Indeed, knockdown of FMNL1 was also identified in our track-etch membrane-based screens and may more specifically support movement as cells squeeze through small pores. In the 3D context, formin-like 1 may help to support more persistent movement as cells move through the complex fibrous network.

In contrast to formin-like 1, we find coronin 1A predominantly colocalized with the lamellipodial filamentous actin structures at the cell front, though more diffuse protein localization was also observed at the cell rear (Fig. 6d and Supplementary Fig. 6b). Coronin 1A localization to the lamellipodial projections is consistent with prior characterization of cells migrating on a 2D surface[70,73,74], while the rear-localized protein may relate to a role in actin turnover

and disassembly[68]. Here, the lack of change in migratory persistence following CORO1A knockdown may relate to a more general disruption of actin cytoskeleton dynamics, rather than alterations to how cells move through the collagen ECM, though further work will be needed to clarify this point.

## CRISPRi screens of cell migration provide a rich resource for studying rapidly migrating cells

Beyond the genes noted thus far across our migration screens, we identified a variety of additional genes with expected roles in actomyosin-based migration. For example, we identified the β subunit of the filamentous-actin capping protein CapZ (CAPZB), which is known to cap actin filaments at their barbed ends[75] and the adenylyl cyclase-associated protein 1 (CAP1), a regulatory protein which facilitates cofilin-driven actin filament turnover and may also interact with talin 1[76]. With respect to myosin contraction at the cell rear, knockdown of RhoA, a key regulator of myosin activity, significantly perturbed migration across all migration screens. Among modulators of Rho GTPases, knockdown of ARHGAP30 was among the most significant perturbations in our migration screens and has been reported to negatively regulate activity of RhoA and Rac1 by enhancing GTP hydrolysis[77,78].

Our screens also identified many other genes that have less obvious roles in cell migration (complete gene lists provided in Supplementary Data 5–7). To explore our migration data sets more broadly, we applied pathway enrichment analysis to the results of our cell migration screens (Fig. 7a). Combining this with our exploration of the data thus far, in Fig. 7b we provide a summary of genes identified across our cell migration assays.

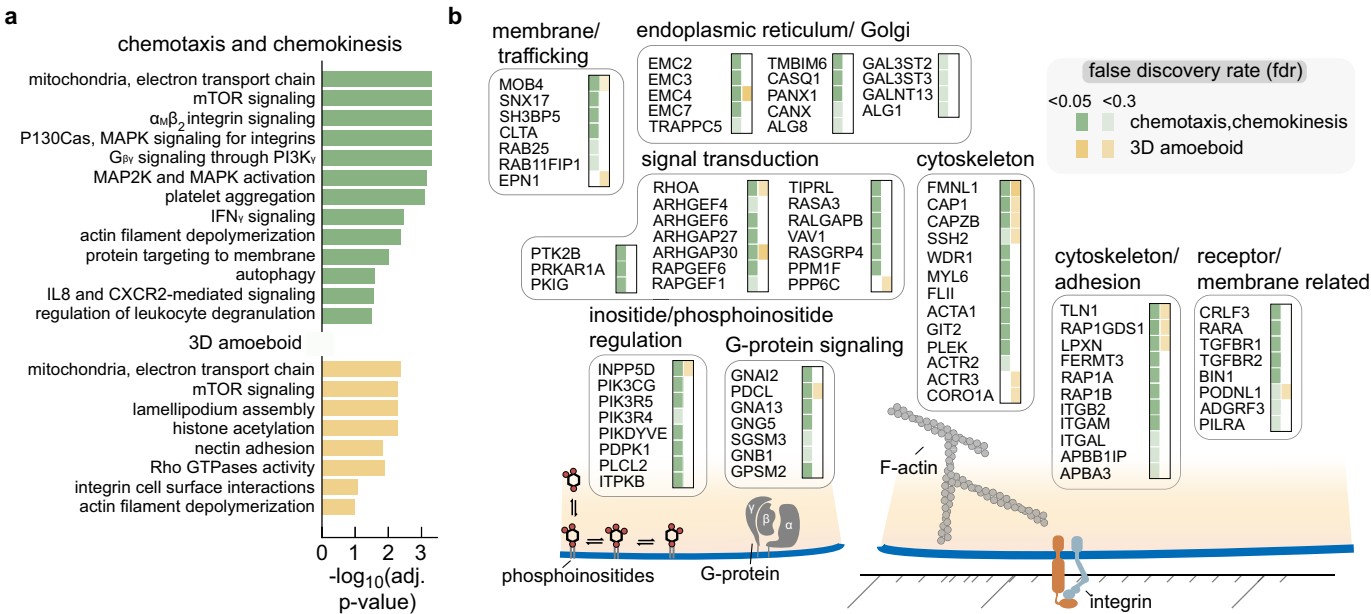

**Fig. 7 | Summary of pathways and genes identified across cell migration CRISPRi screens. a** Pathways enriched in cell migration screens. Since the majority of gene knockdowns lead to poorer migratory phenotypes (i.e., negative $\log_2$ fold-changes), disruption of the noted pathways are associated with poorer migratory success. Due to the correlation across the chemotaxis and chemokinesis screens, their data was combined in this analysis (green). Pathways enriched in the 3D amoeboid migration screen are shown in yellow. $p$ values estimate the statistical significance of gene set enrichment, calculated using a one-sided permutation test and adjusted for multiple comparisons using the Benjamini–Hochberg procedure.

**b** Summary of the genes identified across the cell migration screens. Genes were identified from the collated chemotaxis/chemokinesis screens (green) and 3D amoeboid screen (yellow), with the shading intensity indicating the false discovery rate threshold that each gene fell into (adjusted $p < 0.05$ or $<0.3$). Empty column entries (i.e. white entries) indicate that the gene was not identified as significant. Genes associated with transcription, translation, and gene regulation, or genes that perturbed the processes of either proliferation or differentiation with absolute $\log_2$ fold-changes values larger than 0.7 were excluded.

## Differential sensitivity to protein trafficking machinery and integrin expression is observed across all cell migration assays

Among the candidate genes not obviously associated with cytoskeletal function was VPS29 (Fig. 5e, f), a component of the retromer and retriever complexes. These complexes recycle transmembrane proteins from endosomes back to the trans-Golgi network and the plasma membrane, respectively[79,80]. Upon further examination, additional subunits of both the retromer and retriever complexes were identified as significant hits in our migration screens (Fig. 8a, left). Among other proteins involved in protein trafficking, we also identified components of the HOPS and CORVET complexes (Fig. 8a, right), which are specifically involved in endosomal–lysosomal protein trafficking[81] and may also influence mTORC1 signaling[49].

We considered the hypothesis that the perturbations affecting protein trafficking might be altering integrin recycling and degradation[82]. In further support of this, sorting nexin 17 (SNX17) was also identified in our chemotaxis and chemokinesis screens. This protein binds to β integrins in conjunction with the retriever complex, recycling integrins back to the plasma membrane[83]. To test whether integrin expression was altered when genes associated with membrane recycling were knocked down, we generated additional cell lines with sgRNAs targeting VPS29 and SNX17 and measured cell surface expression of $\alpha_M\beta_2$ integrins (CD11b and CD18, for integrin $\alpha_M$ and $\beta_2$, respectively) using flow cytometry. As a positive control, we found that the ITGB2 knockdown cell line showed a near complete loss of $\beta_2$ integrin expression, as expected (Fig. 8b, right). Only assembled heterodimer $\alpha\beta$ integrin pairs are expected to be stably localized at the cell surface[84], and consistent with this, these cells also exhibit a near-complete loss of integrin $\alpha_M$ (Fig. 8b, left). VPS29 and SNX17 knockdown lines had a moderate drop in $\alpha_M\beta_2$ integrin expression relative to our control sgRNA (Fig. 8b), with measurable decreases in the expression levels of both subunits. These findings show that the

migration defects associated with perturbations to the membrane recycling pathway may be due, at least in part, to disruptions in integrin surface presentation.

Interestingly, although 3D amoeboid migration was insensitive to knockdown of $\alpha_M\beta_2$ integrin (Fig. 6b), several genes among the protein trafficking complexes, including VPS29, still disrupted cell migration when knocked down in our 3D migration screen (Fig. 8a). Although disruption of these protein complexes may have pervasive effects beyond altering integrin expression[28], we wanted to explore the role of integrins more comprehensively across our cell migration screens. Notably, in our 3D amoeboid screen, sgRNA targeting ITGA1 led to a significant perturbation to cell migration (Fig. 8c). Integrin heterodimers containing the integrin $\alpha_1$ subunit are known to bind collagen[85], which was the primary ECM component in this screen. We validated this finding by constructing an individual cell line with sgRNA targeting ITGA1 and tracked its migration in 3D. In contrast to ITGB2 knockdown, which only affected 2D migration, our line targeting ITGA1 showed a modest reduction in cell speed and a significant drop in cellular persistence during 3D migration (Fig. 8d). In summary, we find that protein trafficking, and related alterations in integrin cell-surface expression, can substantially alter behavior of neutrophils across both 2D and 3D cell migration contexts.

## Discussion
The rapid migratory characteristics of neutrophils and their early role in our innate immune response to infection or wounds make them an important cell type to consider in the context of cell migration. In this work, we have demonstrated the use of pooled genome-scale gene perturbations using CRISPRi gene knockdown to study proliferation, differentiation, and cell migration in human HL-60 neutrophils. This was made possible through the additional development of scalable assays that effectively separate cells based on their migratory capabilities.

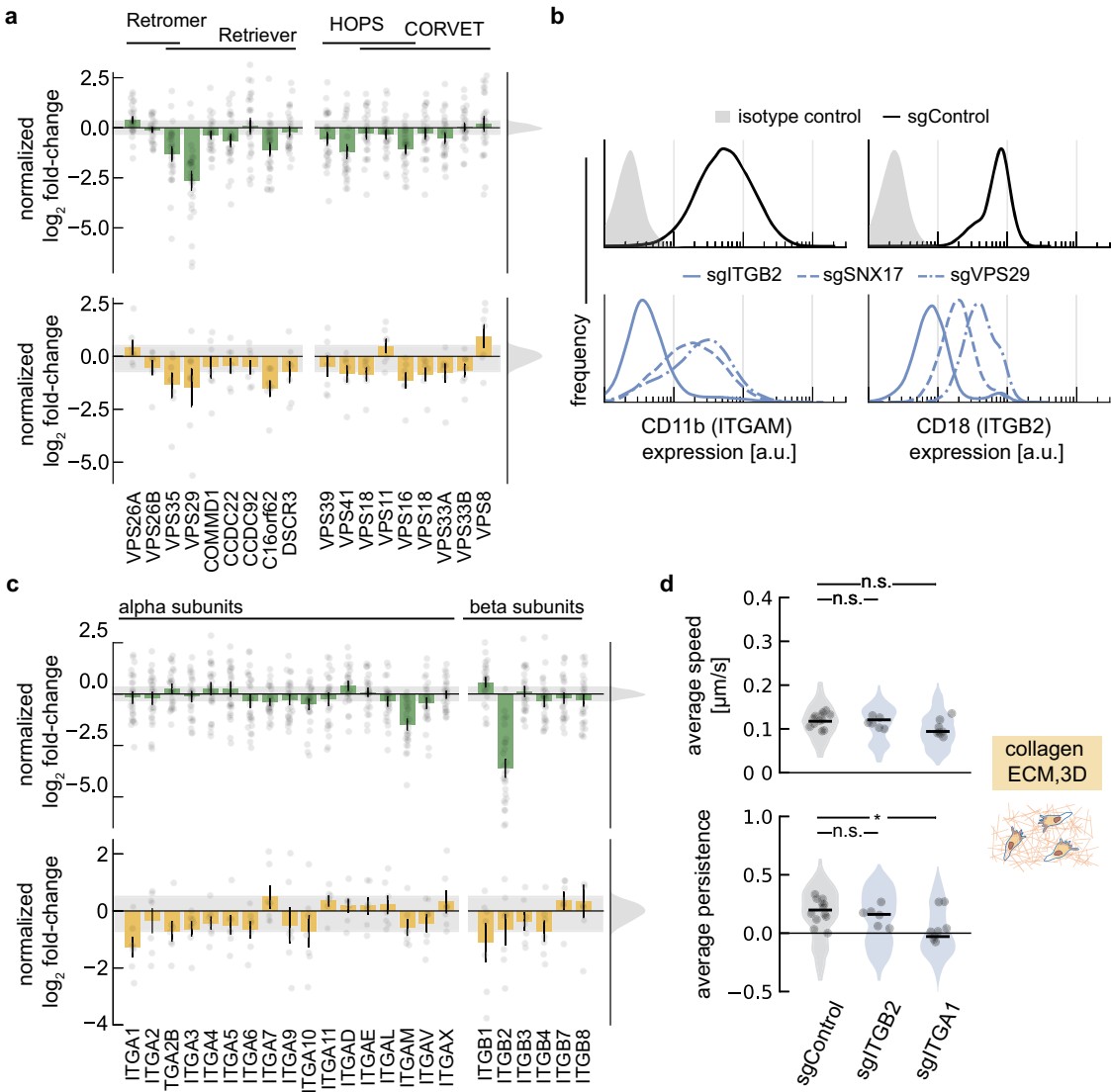

**Fig. 8 | 2D and 3D migration show context specific sensitivities to integrin expression and recycling. a** Summary of normalized log₂ fold-changes of the most significant sgRNA for protein trafficking genes in the chemotaxis and chemokinesis screens (green), and the 3D amoeboid screen (yellow). Error bars represent mean values +/− SEM (green: n = 28 measurements from 14 independent experiments; yellow: n = 7 measurements from 6 independent experiments). Shared gene products for the protein complexes (retromer/ retriever and HOPS/CORVET) are indicated by a solid black line. The histograms and shaded region identify the distribution of the control sgRNAs. **b** Immunofluorescence flow cytometry of CD11b (ITGAM gene; left column) and CD18 (ITGB2 gene; right column). Histograms show surface distribution in control sgRNA (black, solid), ITGB2 (blue, solid), SNX17 (blue, dashed), VPS29 (blue, dash-dot). Shaded histograms indicate cellular autofluorescence from a non-targeting isotype control antibody. **c** Summary of normalized log₂ fold-changes of the most significant sgRNA for integrin genes in

the chemotaxis and chemokinesis screens (green), and the amoeboid screen (yellow). Integrin genes whose transcription is not detected in HL-60 cells[28] were excluded. Error bars represent mean values +/− SEM (green: n = 28 measurements from 14 independent experiments; yellow: n = 7 measurements from 6 independent experiments). The histogram and shaded region identify the distribution of the control sgRNAs. **d** Characterization of cell migration phenotypes in integrin knockdown lines. Speed was calculated by tracking cell nuclei during migration in a collagen ECM. Persistence was inferred from the cell velocity data as described by Metzner et al. (see "Methods"). Individual data points represent mean values for cells across a single field of view, with the shaded regions showing the distribution of all measurements. Measurements represent experiments performed over 2–3 days, acquired across 10 (sgControl), 4 (sgITGB2), and 5 (sgITGA1) fields of view. A two-sided Mann-Whitney U test found the persistence of the ITGA1 knockdown line differed from control cells (*p = 0.002).

With roughly 10¹¹ neutrophils produced in the bone marrow each day[1], any process that impacts neutrophil abundance will severely influence their protective capabilities. As such, we chose to identify genes relevant to differentiation by looking at changes in cell abundance between differentiated and undifferentiated cells following gene knockdown. While some of the results from our screen of differentiation may reflect specific genetic sensitivities of the HL-60 leukemia cell line and our differentiation protocol, we were encouraged by the identification of key transcriptional regulatory genes, including CEBPA, CEBPE, and SPI1, which are known to be involved in

neutrophil differentiation. Beyond these genes, we found that perturbations along the FLCN-RagC/D signaling axis of mTORC1 signaling impacts differentiation and survival, and also dramatically altered cells' migratory phenotypes. We hypothesize that these effects proceed via a non-canonical mechanism of mTORC1 regulation that has only recently begun to be elucidated[49,54].

Intriguingly, we found substantial overlap of enriched genes when comparing our results on differentiation of HL-60 cells with those from a differentiation screen for exit from pluripotency in mouse embryonic stem cells[42,43]. Pathways regulating stem cell renewal and differentiation

share substantial overlap with stress-response pathways[86] and our observations suggest these processes continue to be important as mammalian cells change their proliferative status during myeloid differentiation. In the context of myeloid cells, terminal cell differentiation would provide a useful strategy by the host to minimize the propagation of damaged or cancerous cells, and our results appear particularly relevant to differentiation therapy as a treatment for patients with acute promyelocytic leukemia[35]. While the mechanism behind DMSO-mediated differentiation of myeloid precursors remains poorly understood[19], DMSO can alter membrane permeability and impair mitochondrial function[87,88] and may act as a chemical insult that drives cell differentiation by impairing cellular homeostasis. Mitochondria, in particular, are hubs of metabolic signaling that produce molecules that modulate cellular function, gene expression, and can alter differentiation state[89,90].

Among our cell migration screens, the most striking difference between the track-etch membrane transmigration screens for chemotaxis and chemokinesis as compared to the screens for efficient migration in 3D extracellular matrices was the importance of cell-substrate adhesion. In particular, genes associated with inside-out $\alpha_M\beta_2$ integrin signaling dramatically reduced migration success in our screens using track-etch membranes. These results are especially relevant to the family of leukocyte adhesion deficiency (LAD) disorders, where neutrophils are unable to effectively extravasate from blood vessels into tissue and mount an immune response following infection[91]. Notably, both ITGB2 and FERMT3 are hits in our screens that are also single-gene defects causing the LAD1 and LAD3 subtypes of this disorder, respectively[92,93]. The majority of these adhesion-related genes were not important for 3D amoeboid migration, highlighting the value in using multiple assays to probe cellular function in different, but related contexts. This is also consistent with prior work showing that this type of cell migration is largely integrin-independent[32,67]. However, our 3D screen did allow us to identify a specific role for integrin $\alpha_1$ (ITGA1), where we found that knockdown led to cells with poorer persistence when monitoring single cells migrating in a 3D collagen matrix.

We observed a strong, indeed a near-perfect, correlation between the $\log_2$ fold-change values between chemotaxis and chemokinesis track-etch membrane experiments. As a first attempt to compare these two processes genome-wide, it was surprising to find a lack of any notable enrichment for any genes between the two assays, suggesting that there are no major genetic differences or distinct molecular pathways required for directed as opposed to random neutrophil migration. This was true even though our chemotaxis assay resulted in substantially more cells migrating to the bottom reservoir than the chemokinesis assay over equivalent time frames. This result speaks to the ability of neutrophils, as well as more disparate motile cell types like fish keratocytes[94], to spontaneously polarize their migratory machinery in the absence of any asymmetric spatial cue[29,95,96]. Chemotaxis, at least in the context of serum stimulation as explored here, likely acts to more efficiently guide the movement of cells through the pores of the track-etch membrane, but our results indicate that the underlying molecular mechanisms of directed and spontaneous cell motility are essentially identical. For more specific chemoattractants like fMLF, we would still expect an increase in migratory activity during both chemokinesis and chemotaxis[30], and we hypothesize that the similarity between chemotaxis and chemokinesis may be more broadly applicable.

In summary, our data provides a valuable resource for future study of proliferation, differentiation, and context-dependent cell migration of rapidly migrating neutrophils. Further experimental adjustments may provide additional insights into cell migration. For example, in our 3D amoeboid screen, changes could be made to the ECM composition and density[97] or alternative spatial gradients could be implemented[98,99]. These alternative experimental paradigms could be used to yield new insights into other modes of cell migration like durotaxis (gradients in ECM rigidity)[100], haptotaxis (gradients in substrate composition)[101], or galvanotaxis (directional response to electrical cues)[102,103].

## Methods

### Cell culture and neutrophil differentiation

Undifferentiated HL-60 cells (uHL-60) were a generous gift from the lab of Dr. Orion Weiner. These cells were cultured and differentiated into neutrophil-like cells (dHL-60) as previously described[22,104]. Briefly, cells were maintained at 37 °C and 5% CO2, cultured in RPMI 1640 medium containing L-glutamine and 25 mM HEPES 1640 (Gibco #22400089) supplemented with 10% heat-inactivated fetal bovine serum (hiFBS) (Gemini Bio Products #900–108), and 100 U/mL penicillin, 10 µg/mL streptomycin, and 0.25 µg/mL Amphotericin B (Gibco #15240). Differentiated HL-60 cells (dHL-60) were generated by incubating cells in media containing 1.57% Dimethyl Sulfoxide (DMSO, Sigma, #D2650). Here, confluent cells (approx. $1-1.5 \times 10^6$ cells/mL) were diluted by adding two volumes of additional media and DMSO. The culture media was replenished with fresh media, including DMSO, three days following the initiation of differentiation. Except for where noted otherwise, dHL-60 cells were used in cell migration assays five days following the initiation of differentiation. Experiments involving rapamycin treatment used rapamycin (Thermo Scientific Chemicals #AAJ62473MF).

### CRISPRi pooled library construction

All genomic integrations involved lentiviral transduction. uHL-60 cells expressing dCas9-KRAB were first generated and this cell line was used for all subsequent work (sgRNA genome-wide libraries and individual sgRNA targeting cell lines).

To achieve reliable gene knockdown in uHL-60 cells, we used dCas9-KRAB linked by a proteolysis-resistant 80 amino acid XTEN linker[26,27], driven by an EF1α promoter that was placed downstream of a minimal-ubiquitous chromatin opening (UCOE) element to prevent gene silencing[26,105]. The dCas9-KRAB construct was based on a construct originally gifted by Dr. Marco Jost and Dr. Jonathan Weissman, but modified to include blasticidin resistance (pHR-UCOE-Ef1a-dCas9-HA-2xNLS-XTEN80-KRAB-P2A-Bls).

The sgRNA library was previously reported in Sanson et al. (Dolcetto CRISPRi library set A, Addgene #92385). This library contains 57,050 sgRNA, with 3 sgRNA per gene target and 500 non-targeting control sgRNA. For optimal library design, sgRNA were selected based on their position relative to annotated transcription start sites, expected on-target activity, and the presence of off-target matches.

For large-scale lentivirus production (dCas9 construct or pooled sgRNA library), 15 µg transfer plasmid, 18.5 µg psPAX2 (Addgene #12260), and 1.85 µg pMD2.G (Addgene #12259) were diluted in 3.5 ml Opti-MEM I reduced-serum media (Gibco #31985070) and then combined with 109 µL TransIT-Lenti Transfection Reagent (Mirus, MIR6600). Following a 10 minute incubation, this mixture was added dropwise to confluent HEK-293T cells (ATCC, CRL-3216) in a T175 flask containing 35 mL DMEM media (Gibco #11965-092) and supplemented with 1 mM sodium pyruvate (Gibco #11360-070). Lentivirus was recovered by collecting media 48 hr later, with centrifugation at $500 \times g$ for 10 min to remove any residual cells and debris. For our dCas9-KRAB construct, we additionally concentrated the lentivirus approximately 60-fold using Lenti-X Concentrator (Takara Bio Inc., #631231).

For small-scale lentivirus production (individual sgRNAs), lentivirus was prepared in 6-well tissue culture plates. Here 1 µg sgRNA transfer plasmid, 1 µg psPAX2, and 0.1 µg pMD2.G were diluted in 200 µL Opti-MEM I reduced-serum media and combined with 6 ul transIT. Following a 10 minute incubation, this mixture was added dropwise to confluent HEK-293T cells, with lentivirus collected as noted above.

## Construction of individual sgRNA plasmids for CRISPRi

Individual sgRNA plasmids were constructed for the generation of stable CRISPRi knockdown cell lines using sgRNA identified from the genome-wide CRISPRi screens. These were cloned into the same base vector pXPR_050 plasmid (Addgene #96925) as the pooled library, as previously described[10]. Briefly, pXPR_050 was first linearized using the restriction enzyme BsmBI (New England Biolabs, #R0739S, which includes NEB buffer 3.1). Here, 20 µg of pXPR_050, 20 µL NEB buffer 3.1, and 10 µL BsmBI were combined for a 200 µL reaction and incubated for 5 hours at 55 °C. The resulting linear pXPR_050 DNA was gel extracted using the QIAquick gel extraction kit (Qiagen, #28704) and resuspended in TE buffer (10 mM Tris·Cl, pH 8.0; 1 mM EDTA) to a concentration of 10 ng/µL. sgRNA inserted were generated by annealing complementary oligonucleotides with DNA overhangs compatible for ligation with the BsmBI-digested pXPR_050 DNA. The oligonucleotides were purchased from Integrated DNA Technologies (Coralville, IA) as described below and annealed by combining 1.5 µL of each forward and reverse oligonucleotide (stock concentration of 50 µM in water), 5 µL NEB buffer 3.1, and 42 µL water. The mixture was first incubated for 5 min at 95 °C and then allowed to cool by lowering the temperature by 5 °C every 5 min until the sample was at room temperature. Finally to ligate the sgRNA insert into the pXPR_050 vector, 1 µL of annealed sgRNA insert was combined with 20 ng of the BsmBI-digested pXPR_050 DNA and ligated using T4 ligase (New England Biolabs, #M0202S). The ligated DNA product was transformed into NEB Stable Competent *E. coli* (New England Biolabs, #C3040H) following the manufacturer directions and successfully inserted sgRNA were identified by Sanger sequencing (performed by Genewiz from Azenta Life Sciences; Burlington, MA).

Forward oligonucleotides were ordered as 5' CACCG (20 bp sgRNA target sequence)3', while reverse complement oligonucleotides were ordered as 5' AAAC (20 bp reverse complement sgRNA target sequence)C 3'. The forward 20 bp sgRNA target sequences used for individual CRISPRI knockdown cell lines are listed below.

Control sgRNA: AGGGCACCCGGTTCATACGCNGG;
GIT1 sgRNA: GGCGGCGCTTCCGCTCTAACNGG
FMNL1 sgRNA: GCCCCGTCCGTGGGACCGGGNGG
TSC1 sgRNA: GACTGTGAGGTAAACAGCTGNGG
ATIC sgRNA: CTGGGTTCAGGGCGAGCGGGNGG
RICTOR sgRNA: CGGGCTTACCTCGTACTCGGNGG
ITGA1 sgRNA: CGTGTTTAGGCTAAAGTCCANGG
APBBIIP sgRNA: CCTTAGTCCCTCTTGCGTCGNGG
CORO1A sgRNA: ATCTTCAGCGGGCGAGTCCCNGG
VPS29 sgRNA: CGACGGTGGTGGTGACTGAGNGG
SNX17 sgRNA: TGCGGGGACTCGCTGAGCAGNGG
ITGB2 sgRNA: CGGTGTGCTGGAGTCCTCGGNGG
CEBPE sgRNA: GTAGGCGGAGAGGTCAATGGNGG
SPI1 sgRNA: CCCAGGGCTCCTGTAGCTCANGG
ARHGAP30 sgRNA: CAGGACACAATTTCTTGCCANGG
FLCN sgRNA: GCCCGGGTTCAGGCTCTCAGNGG
TLN1 sgRNA: GGGCGACCCGAGAAGCGGCGNGG
LAMTOR1 sgRNA: GCTGCTGTAGCAGCACCCCANGG.

## CRISPRi cell line construction

The dCa9-KRAB and sgRNAs constructs were integrated into uHL-60 cells using a lentivirus spinoculation protocol. Briefly, lentivirus was added to 1 mL cells (1 × 10⁶ cells/mL) and polybrene reagent (final concentration of 1 µg/mL) in 24-well tissue culture plates. Cells were spun at 1000 × *g* for 2 h at 33 °C. Virus was removed and cells were placed in an incubator for 2 days prior to antibiotic selection for 6 days (dCas9-KRAB: blasticidin 10 µg/mL; sgRNA constructs: puromycin 1 µg/mL).

For CRISPRi sgRNA library preparations, lentiviral titers were estimated by titrating lentivirus over a range of volumes (0 µL, 75 µL, 150 µL, 300 µL, 500 µL, and 800 µL) with 1 × 10⁶ cells in a total of 1 mL

per well of a 24-well tissue culture plate, using the spinoculation protocol noted above. Two days post-transduction, cells were split into two groups, with one placed under puromycin selection. After 5 days, cells were counted for viability. A viral dose that led to a 12.5% transduction efficiency was used for subsequent pooled library work. This low efficiency was targeted to ensure most cells only received one sgRNA integration[10]. For library work, roughly 230 million cells were transduced, targeting a final number of roughly 30 million successful sgRNA integrations following puromycin selection (or about 500 cells per sgRNA). Cells were maintained across multiple T175 tissue culture flasks with 35 mL of media.

Knockdown was confirmed in the ITGB2 knockdown line by flow cytometry immunofluorescence, in the FLCN, LAMTOR1, SPI1, and ATIC knockdown lines by RNA-seq, and in the FMNL1 knockdown line by immunofluorescence microscopy.

## Genome-wide CRISPRi assays

**Overview of cell collection and experimental replicates.** For CRISPRi proliferation drop-out screens, genomic DNA (gDNA) was collected from uHL-60 cells at two time points, separated by 6 days of proliferation in T175 tissue culture flasks. Each cell preparation was pelleted by centrifugation and frozen for later genomic DNA isolation. Results from the proliferation screen represent averages across four independently prepared genome-wide CRISPRi libraries.

For CRISPRi differentiation drop-out screens, gDNA was collected in uHL-60 and in dHL-60 cells 5 days following the initiation of differentiation in 15 cm tissue culture dishes. Results from differentiation screens represent averages across four independently prepared genome-wide CRISPRi libraries, each performed twice (8 experiments total).

For cell migration experiments, gDNA was collected from three populations: On the day of each experiment, 5-day differentiated dHL-60 cells were collected, with 3×10⁷ cells set aside as a reference sample (3 in Fig. 1b). For chemotaxis and chemokinesis experiments involving track-etch membranes, the other two populations were the fraction of cells that migrated through the membrane (4ii in Fig. 1c) and the fraction of cells that remained on top of the membrane (4i in Fig. 1b). For the amoeboid 3D screen, the other two populations were the fraction of cells that migrated into the fibrin (5ii in Fig. 1c) and the fraction of cells that remained in collagen (5i in Fig. 1c).

Regarding experimental replicates, for assays using track-etch membranes, 6-h time point chemotaxis experiments were performed across four independently prepared genome-wide CRISPRi libraries, with eight experiments in total. For 6-h chemokinesis experiments and all 2-h time points (both, chemotaxis and chemokinesis), experiments were performed using two independently prepared genome-wide CRISPRi libraries (two experiments each). For amoeboid experiments, results represent the average across two independently prepared genome-wide CRISPRi libraries, each performed three times (six experiments total).

Chemotaxis and chemokinesis cell migration experiments with track-etch membranes compared the number of cells that migrated through the pores (4ii in Fig. 1c) with respect to the reference sample, and those that did not (4i in Fig. 1b) with respect to the reference sample. For the 3D amoeboid migration screen we examined the fraction of cells that migrated into the fibrin (5ii in Fig. 1c) with respect to the reference sample, and those that remained in the collagen (5i in Fig. 1c) with respect to the reference sample. This resulted in two separate measurements per migration experiment, except for several samples where the sgRNA did not PCR amplify properly from the gDNA and was therefore not sequenced.

**Removal of cell debris and dead cells prior to cell migration assays.** Cellular debris and dead cells were removed from the dHL-60 cell suspensions using density gradient centrifugation. Pooled CRISPRi

libraries were differentiated in 15 cm dishes (55 ml cell culture per dish) and eight plates were combined for a single preparation. Cells were first spun down (10 min at $300 \times g$), resuspended in 10 mL Poly-morphPrep (Cosmo Bio USA #AXS1114683) and added to the bottom of a 50 mL conical tube. Using a transfer pipette, 15 mL of 3:1 Poly-morphPrep: RPMI media + 10% hiFBS was gently layered on top by dispensing along the walls of the tube. This was followed by layering another 14 mL of RPMI media + 10% hiFBS. Cells were centrifuged at $700 \times g$ for 30 min with the centrifuge acceleration and brake set to half-speed. Live dHL-60 cells were collected between the RPMI media and the 3:1 PolymorphPrep. RPMI media layers were diluted with one volume of RPMI media + 10% hiFBS, and spun down once more for 10 min at $300 \times g$. Finally, cells were resuspended in 10 mL RPMI media and counted using a BD Accuri C6 flow cytometer (live cells identified by their forward scatter and side scatter).

**Cell migration assays: chemotaxis and chemokinesis using track-etch membranes.** Chemotaxis and chemokinesis transwell migration assays[106] used track-etch membranes with $3 \, \mu m$ pore sizes (6-well plates with 24.5 mm diameter inserts; Corning, #3414). For each experiment, 24 million cells were distributed across four 6-well plates. For each track-etch membrane insert, one million cells were diluted in 1.5 mL media and added to the top of the track-etch membrane. Note that for chemotaxis experiments, cells were purified and resuspended in RPMI media without hiFBS. To the bottom reservoir, 2.6 mL RPMI media with 10% hiFBS was added and the plates were carefully moved to a 37 °C incubator. Following incubation for the required time (2 and 6 h time points), the track-etch inserts were separated from the bottom reservoir. To ensure more complete recovery of the migratory cells, the bottom side of the track-etch membrane was gently scraped using a cell-scraper (Celltreat, #229310) to dislodge any cells remaining on the membrane surface. The migratory cells (bottom reservoir) and remaining cells (top reservoir) were separately collected by centrifugation. Following an additional wash with 1 mL PBS, cells were pelleted and frozen at −80 °C for later genomic DNA extraction.

Note that in a second set of chemotaxis experiments (6-h time point; four of the replicates), custom devices were fabricated to house larger 49 mm track-etch membranes ($3 \, \mu m$ pore size, Sigma #TSTP04700). Similar cell densities were targeted as the experiment using multi-well plates above.

**Cell migration assays: amoeboid 3D using collagen and fibrin ECM.** Cells were seeded into a multi-layer system of collagen (rat-tail collagen used throughout; ThermoFisher # A1048301) and fibrin as shown in Fig. 1c. Briefly, these were prepared by first creating a ~50 μm layer of collagen on top of 25 mm glass coverslips, seeding dHL-60 cells in another layer of collagen ~150 μm thick, and then overlaying this with fibrin ECM. Each genome-wide CRISPRi screen involved our pooled dHL-60 library spread across 32 coverslips with $1 \times 10^6$ cells added to each coverslip.

The glass coverslips were first surface modified to support an adhered layer of collagen ECM using a silane treatment[107]. A 2% ami-nosilane solution was first prepared in 95% ethanol/5% water and incubated for 5 min to allow silanol formation. Coverslips were then immersed in the solution for 10 min, rinsed with 100% ethanol, and then cured on a hot plate heated to 110 °C for 5–10 min. The coverslips were then immersed in 0.25% glutaraldehyde for 15 min and then rinsed in water for 5 min. This 5 min wash was repeated two more times. We then prepared the initial collagen layer by pipetting 22.6 μL of a 1.5 mg/mL collagen mixture (for 3 mL: 1.5 mL 3 mg/mL collagen, 376 μL 0.1 M NaOH, 210 μL 10× PBS, and 923 μL PBS) onto a 15 cm plastic tissue culture dish and then placing a coverslip on top, causing the mixture to spread across the entire coverslip. Approximately 16 coverslips were prepared inside a single 15 cm tissue culture dish. The dish was placed in an incubator at 37 °C for 18 min to gel.

Coverslips containing the initial layer of collagen were carefully removed using tweezers and flipped collagen side up to allow dHL-60 cells to be seeded. Here, a 1 mg/mL collagen mixture was prepared and mixed with dHL-60 cells (recipe for 1.2 ml: 400 μL 3 mg/mL collagen, 100 μL 0.1 M NaOH, 55 μL 10x PBS, 525 μL PBS, and 120 μL hiFBS). For each collagen treated coverslip, $3 \times 11.3 \, \mu L$ aliquots were pipetted on top of the initial collagen layer, which wetted and spread across the initial collagen layer. Note that the initial coverslip and collagen layer will begin to dry out while inside a biosafety cabinet due to the air flow, so this second collagen mixture must be added relatively quickly to ensure proper spreading of this second layer. The coverslips were again moved to an incubator at 37 °C for 15 min to gel, placed inside a closed tissue culture dish containing a wetted kimwipe to minimize evaporation. Note that during this time, prior to gelling, cells will settle down to the initial collagen layer.

The final layer, composed of fibrin ECM[108], was then prepared on top of the collagen. Here, a 1 mg/mL fibrin ECM was generated by mixing 1 μL thrombin (100 U/ml, Sigma-Aldrich #T1063-250UN) per 1 mL fibrinogen at 1.5 mg/mL (plasminogen-Depleted from human plasma, Sigma-Aldrich #341578) in 1× Hanks' buffered salt solution (HBSS; ThermoFisher Scientific, Gibco #14-065-056). This was carefully added on top of the collagen by slowly pipetting near the edge of the coverslip. The fibrin ECM will begin to gel immediately, but the coverslips were further incubated at 37 °C for 18 min to complete the gelling process. Finally, the multi-layered gels were covered with RPMI media containing 10% hiFBS and incubated for 9 h for dHL-60 cells to migrate.

Recovery of the most migratory dHL-60 cells from the fibrin ECM were obtained by first incubating coverslip/gels with nattokinase, which specifically degrades fibrin[34]. Here, the RPMI media was first aspirated from the 15 cm dish and 25 ml PBS containing 2.1 mg/mL nattokinase (Japan Bio Science Laboratory USA inc.) and 0.5 M EDTA were added, incubating at 37 °C for 40 min. The coverslips and remaining collagen layer was then removed, allowing recovery of the released dHL-60 cells. The remaining coverslips + collagen were rinsed with 25 mL PBS + 10% hiFBS prior to scraping the collagen together for later gDNA extraction.

**Quantification of sgRNA from CRISPRi libraries.** Genomic DNA (gDNA) was isolated using QIAamp DNA Blood Maxi ($3 \times 10^7 – 1 \times 10^8$ cells) or Midi ($5 \times 10^6 – 3 \times 10^7$ cells) kits following protocol directions (Qiagen, #51192 and #51183). gDNA precipitation was then used to concentrate the DNA. Briefly, salt concentration was adjusted to a 0.3 M concentration of ammonium acetate, pH 5.2 and 0.7 volumes of isopropanol were added. Samples were centrifuged for 15 minutes at $12,500 \times g$, 4 °C. Following a decant of the supernatant, the gDNA was washed with 10 mL 70% ethanol and spun at $12,500 \times g$ for 10 min, 4 °C. The samples were washed in another 750 μL 70% ethanol, spun at $12,500 \times g$ for 10 min, 4 °C, and decanted. The pellets were allowed to air-dry prior to resuspending them in water. The gDNA concentrations and purity were determined by UV spectroscopy.

The sgRNA sequences from each gDNA sample were PCR amplified for sequencing following protocols provided by the Broad Institute's Genetic Perturbation Platform. Briefly, gDNA samples were split across multiple PCR reactions, with 10 μg gDNA added per 100 μL reaction: 10 μL 10× Titanium Taq PCR buffer, 8 μL dNTP, 5 μL DMSO, 0.5 μL 100 μM P5 Illumina sequencing primer, 10 μL 5 μM P7 barcoded Illumina sequencing primer, and 1.5 μL Titanium Taq polymerase (Takara,# 639242). The following thermocycler conditions were used: 95 °C (5 min), 28 rounds of 95 °C (30 s)–53 °C (30 s)–72 °C (20 s), and a final elongation at 72 °C for 10 min. PCR products (expected size of ~360 bp) were gel extracted using the QIAquick gel extraction kit (Qiagen, #28704) following protocol directions. After elution, samples were further cleaned up using isopropanol precipitation. Here, 50 μL PCR DNA samples were combined with 4 μL 5 M NaCl, 1 μL GlycoBlue

coprecipitant (ThermoFisher Scientific Technologies,# AM9515), and 55 µL isopropanol. Samples were incubated for 30 min and then centrifuged at 15,000 × g for 30 min. The resulting pellet was washed twice with 70% ice-cold ethanol and resuspended in 25 µL of Tris-EDTA buffer. Illumina 150 bp paired-end sequencing was performed by Novogene Corp. (Sacramento, CA).

Sequence reads were quality filtered by removal of reads with poor sequencing quality, and reads were associated back to their initial samples based on an 8 bp barcode sequence included in the P7 PCR primer. The 20 bp sgRNA sequences were identified and mapped to gene targets using a reference file for the genome-wide CRISPRi library[10].

Log2 fold-change values were calculated from the sequencing counts between two sets of samples. The specific comparison sets made for each screen are described in section "Overview of cell collection and experimental replicates." Note that each cell migration assay resulted in two log2 fold-changes measurements, since two populations of cells were collected and each compared to a reference set of sgRNA. Since we expect these two measurements to be inversely correlated, the log2 fold-change values from the less-migratory population were multiplied by −1. For example, if gene knockdown resulted in a negative log2 fold-change in the bottom reservoir of our chemotaxis track-etch membrane experiment, we would expect a positive log2 fold-change for that sgRNA in the upper reservoir. The multiplication by −1 allowed the two sets of log2 fold-change values to be compared directly and we averaged across all such measurements.

Reported Log2 fold-changes represent averages across median-normalized replicate measurements from the multiple experiments performed. Here, a pseudocount of 32 was added to the sgRNA counts to minimize erroneously large fold-change values in cases of low library representation[109]. For the differentiation screen and cell migration screens, log2 fold-changes were also scaled to have unit variance prior to averaging across individual experiments[110]. p Values were determined by performing permutation tests[111] between the calculated fold-change values for each gene target and our set of ~500 control sgRNA fold-change values. Adjusted p values for multiple comparisons were determined using the Benjamini−Hochberg procedure[112].

### Gene set enrichment analysis

Enrichment analysis was performed using the GSEA software version 4.1.0[113] following recommended parameters[36]. A subset of significant gene ontology terms are shown. Statistical analyses were performed by the GSEA software.

### Image acquisition

All microscopy-based image acquisition was performed using setups operated by MicroManager (v. 2.0)[114]. Details of the microscope configurations are provided below for each of the cell migration assays.

### Cell migration assays using individual sgRNA CRISPRi lines

**2D migration on fibronectin coverslips.** Ibidi µ-Slide I slides (#80106, IbiTreat) were incubated with 10 µg/mL fibronectin (from human plasma, #2006, Sigma-Aldrich Inc.) in PBS at room temperature for 1.5 h. The channel slides were then washed once with 1 mL RPMI media containing 10% hiFBS, once with Leibovitz's L-15 Medium (Thermo-Fisher Scientific, Gibco #11415064), and then the media was removed from the reservoirs of the channel slide. Separately, $2 \times 10^5$ dHL-60 cells were collected, resuspended in 1 mL L-15 + 10% hiFBS media containing 1 µg/mL DNA stain Hoechst 33324 and incubated at 37 °C for ~15 min. Cells were then spun down, resuspended in 200 µL L-15 + 10% hiFBS media, and pipetted into one side of the channel slide inlet. Following a 30 min incubation at 37 °C to allow cells to settle and adhere, the slide was rinsed with three washes of L-15 + 10% hiFBS media to remove any remaining floating cells. We note that the switch

from RPMI to L-15 was made to avoid the need for $CO_2$ in our microscope system for pH buffering.

Cells were imaged for nuclei tracking at 37 °C on an inverted microscope (Nikon Ti Eclipse), using a ×20 objective lens (Nikon ×20 0.75 NA plan apo phase contrast), with sequential phase contrast and epifluorescence illumination through a standard DAPI filter set. For each sample, a 30 min time-lapse movie was acquired with 60 s intervals. Individual experiments were performed over 2–3 days, with five different fields of view during each acquisition. For the higher-magnification still images, cells were imaged with a ×100 objective lens (Nikon ×100 1.45NA plan apo) with an additional ×1.5 intermediate magnification. Images were captured on an iXon EMCCD camera (Andor).

**2D migration using agarose overlay and fMLF photo-uncaging.** To assess chemotaxis, we stimulated dHL-60 cells with N-formyl-methionine-alanine-phenylalanine (fMLF) as previously described[30]. Briefly, to reduce adhesion, glass bottom 96 well plates (Cellvis, #P96-1.5H-N) were treated with 1% BSA (Millipore Sigma, #A7979) in water for 15 min followed by two washes with water then dried overnight at 37 °C incubator with lid slightly ajar. Approximately 1000 dHL-60 cells labeled with Celltracker Orange (ThermoFisher, #C2927) were plated in each well, in a 5 µL drop of modified L-15 + 2% hiFBS media. Cells were allowed to adhere to the glass for 5 min, before a 195 µL layer of 1.5% low-melt agarose solution in L-15 (Goldbio, #A-204-25) was mixed 1:1 with L-15 + 10% hiFBS media (warmed to 37 °C) and overlaid on top. We note that the initial agarose mixture was allowed to cool to approximately 37 °C prior to preparing the final dilution and then used immediately.

The agarose was allowed to solidify at room temperature for 40 min and then the plate was transferred to the 37 °C microscope incubator 40 min prior to imaging. Imaging is done using a Nikon Ti-E inverted microscope controlled by MATLAB via Micromanager, allowing simultaneous automated imaging of multiple wells in groups. An environmental chamber was used to maintain a 37 °C temperature, and wells were imaged at 4X magnification every 30 s using epi-fluorescence illumination with the X-Cite XLED1 LED (Excelitas Technologies, GYX module). The excitation light was filtered using a Chroma ZET561/10 band pass filter (custom ZET561/640x dual laser clean-up filter). A caged UV-sensitive derivative N-nitroveratryl derivative (Nv-fMLF) of fMLF was used at 300 nM final concentration and uncaged on the microscope by exposure to UV light with a filter cube with 350/50 bandpass filter (max intensity around 360-365 nm). The initial gradient was generated with a 1.5 s exposure of UV light and recharged with 100 ms exposure after every frame. Image processing and statistical analyses of chemotaxis were performed using custom MATLAB software (Collins et al.[30]). Note that for each experiment, each cell line was added to 24−32 wells. Each well contained roughly 100 cells, resulting in about 20,000 cells quantified for each cell line across five experiments.

**3D migration in collagen ECM.** Cells were prepared as previously described[104]. Briefly, $2 \times 10^5$ dHL-60 cells were collected, resuspended in 1 mL L-15 + 10% hiFBS media containing 1 µg/mL DNA stain Hoechst 33324 and incubated at 37 °C for ~15 min. During incubation with Hoechst stain, a 200 µL collagen aliquot was prepared: 6.5 µL 10x PBS, 12.5 µL 0.1 M NaOH, 111 µL L-15, and 20 µL hiFBS were combined with 50 µl 3 mg/mL collagen. The cell suspension was spun down and resuspended in the collagen mixture for a final concentration of 0.75 mg/mL collagen, and then added to the channel of an Ibidi µ-Slide I (Ibidi, #80106). After 1 min incubation at room temperature, the channel slide was inverted to help prevent cell sedimentation and incubated at 37 °C for gel formation. After 20 min, the channel slide media reservoirs were filled with 2 mL total L-15 media containing 10% hiFBS and imaged within 30 min to 1.5 h.

 

Cells were imaged at 37 °C on an inverted microscope (Nikon Ti Eclipse) with a ×20 0.75 NA objective lens using sequential phase contrast and epifluorescence illumination through a standard DAPI filter set. For each sample, a 60 min time-lapse movie was acquired at 60 s intervals. A z-stack was acquired over 200 μm with acquisitions every 3 μm. In general experiments were performed over three different days, with two 60 min acquisitions taken each day. Images were captured on an iXon EMCCD camera (Andor).

## Cell tracking and quantification of cell migration characteristics

Cell tracks were extracted from the DNA channel of time-lapse microscopy images using custom code[104] with Python (v. 3.9.13). Briefly, nuclei were first identified using a morphological mean filter with a 50 pixel radial disk structural element and thresholding using the Python package scikit-image (v. 0.19.2)[115]. For each nuclei identified, the z-coordinate was calculated by taking a weighted-intensity average along the z-axis. With cell coordinates in hand, cell trajectories were determined by calculating all possible cell-to-cell displacements between consecutive time points, and then matching cells through minimization of the total displacement across cells. For example, cells whose displacement changed very little between two time points would most likely correspond to the same cell. The cell density in the ECM was kept low enough that individual cell tracks could be easily identified. Cell track information, including position and time, we aggregated into a single table using pandas Python package (v. 1.4.4)[116].

Non-overlapping velocities and cell speeds were calculated using the 30 s (fibronectin-coated coverslips) and 60 s (collagen ECM) frame rate of our image acquisition. To estimate average migratory persistence from each cell trajectory, cell tracks were analyzed using a Bayesian inference algorithm based on a persistence random walk[63]. Specific parameters were chosen empirically to best capture persistence changes in the tracks (inference grid size = 200, pMin = $10^{-5}$, persistence box kernel radius = 2, activity box kernel radius = 2). Persistences were allowed to range from −1 to 1, and activities were allowed to range from 0 to 0.5 μm/s.

Comparisons between knockdown and control cell lines were performed using the two-sided Mann-Whitney U nonparametric test (scipy.stats.mannwhitneyu(), using the SciPy Python package (v. 1.9.1)[117].

## Adhesion assay using individual sgRNA CRISPRi lines

Approximately 400,000 dHL-60 cells were resuspended in 250 μL RPMI media containing 10% hiFBS and placed into wells of a tissue culture treated polystyrene plate (Genesee Scientific, #25-107). Following incubation at 37 °C for 45 min, wells were aspirated to remove non-adherent cells. Control wells were also included for each cell line where cells were not aspirated. One gentle wash was performed by adding 500 μL RPMI media containing 10% hiFBS slowly to the side of each well and then aspirating the media. Cells were collected by dislodging all remaining cells by adding 500 μL and pipetting repeatedly. This was performed twice, resulting in cells resuspended in 1 mL PBS. Cells were counted and the fraction of adhered cells was determined by dividing by the total cell count found in the control wells.

## Immunolabeling for flow cytometry

Live-cell immunofluorescence measurements of cell surface expression for CD11b (integrin $α_M$), CD18 (integrin $β_2$) and formyl peptide receptor FPR1 were performed on a Sony SH800 Cell Sorter. All staining and washes were done with cells suspended in phosphate-buffered saline (PBS) containing 2% hiFBS and 0.1% sodium azide, chilled on ice. For each sample, one million cells were first resuspended in 100 μL buffer containing 5 μL Fc Receptor Blocking Solution (Biolegend, #422302) and incubated for 15 minutes. Cells were then spun down and resuspended in 100 μL of buffer containing fluorescently conjugated antibodies for one hour (5 μL of each antibody per sample). Following staining, the samples were washed three times by resuspending in 300 μL of fresh buffer. Following collection of flow cytometry data,.fcs files were exported and processed using the Python package FlowCytometryTools (v. 0.5.1)[118]. Gating was performed on the forward and side scatter to isolate the population of live cells (Supplementary Fig. 7).

**Antibodies and dilution information.** Fluorophore-conjugated anti-human primary antibodies: BB515 Mouse Anti-Human CD11b (1:20; BD Biosciences, B564517), CD18 Mouse anti-Human, FITC (1:20; BD Biosciences, #B555923), fMLF Receptor Mouse anti-Human, Alexa Fluor 647(1:20; BD Biosciences, #565623), and Alexa Fluor® 647 Mouse Anti-Human CD52 (1:20; BD Biosciences, #563610). Isotype controls: BB515 anti-IgG1, (1:20; BD Biosciences, #B564416), FITC anti-IgG1 (1:20; BD Biosciences, #B550616), and Alexa Fluor 647 anti-IgG1 (1:20; BD Biosciences, #B557714).

## Immunolabeling for Western blots

Whole-cell protein lysates were collected from dHL-60 cells for western blot analysis. For each sample, $5 \times 10^6$ cells were collected, washed twice in ice-cold PBS, and resuspended in 100 μL RIPA lysis buffer (Cell Signaling, #9806) containing a protease and phosphatase inhibitor cocktail (Cell Signaling, #5872). The suspension was incubated on ice for 10 min and vortexed briefly prior to sonication with a bath type sonicator (Diagenode, #B01020001). Sonication was performed on their high power setting at 4°C with five cycles of 30 s on and 30 s off. Following sonication the suspension was spun at $15,000 \times g$ for 10 min at 4°C. Finally each sample was diluted with 4× Laemmli SDS-PAGE sample buffer and heated to 98°C for 5 min.

Samples were run on 7.5% polyacrylamide gels with a protein ladder (Bio-rad, #1610317) and and transferred to nitrocellulose membranes (Bio-rad, #1620233) by semi-dry transfer in buffer 10 mM CAPS pH 11, 10% methanol. Transferred protein was assayed using a reversible total protein stain kit (Pierce, #24580) prior to blocking in Tris-buffered saline with 0.1% Tween 20 detergent (TBST) with 0.2% fish skin gelatin (FSG) for 30 min at room temperature. Protein loading was also assessed by staining the residual protein on the gel using Coomassie stain (0.006% Coomassie R250 with 10% acetic acid). Primary antibodies were diluted in TBST with 0.2% FSG and incubated overnight at 4°C, which were co-stained with an Alexa Fluor 790 Anti-GAPDH antibody for loading control. Blots were washed with TBST for 15 min, with buffer exchanged every 5 min, and then stained with an HRP conjugated secondary antibody diluted in TBST with 0.2% FSG. Following incubation for 60 min at room temperature, the blots were washed for 30 min in TBST, with buffer exchanged every 5 min. The blots were imaged with a digital gel documentation system (Azure c600), allowing for detection of the secondary HRP antibody detected using a chemiluminescence peroxidase substrate kit (Sigma, #CPS-1) and subsequent detection of the GAPDH loading control detected using its laser based infrared detection system. Three to four blots were performed for each sample and analyzed using BioRad ImageLab (v. 1.6).

**Antibodies and dilution information.** Rabbit anti-human primary antibodies: Total S6K (1:1000; Cell Signaling, #2708), p-Thr389 S6K pAb (1:1000; EMD Millipore, #07-018-I), Total Akt (1:1000; Cell Signaling, #4691), p-Ser473 Akt (1:2000; Cell Signaling, #4060), total mTOR (1:1000; Cell Signaling, #2983), p-Ser2448 mTOR (1:1000; Cell Signaling, #5536), Alexa Fluor 790 to GAPDH (1:1000; Abcam loading control, #ab184578). Secondary antibody: HRP-linked anti-rabbit IgG (1:3000; Cell Signaling, #7074 S).

## Immunolabeling for fluorescence microscopy during 3D migration

Immunofluorescence imaging was performed in cells migrating in 3D collagen gels. We began by creating a thin layer of collagen on air

plasma-treated 25 mm glass coverslips (5 min at 200 mTorr treatment; Harrick Plasma #PDC-001). This was achieved by placing a coverslip on top of a 30 µL droplet of 0.75 mg/mL collagen mixture in a glass petri dish, which caused the collagen mixture to spread across the coverslip. The collagen was allowed to gel at 37 °C for 90 min. Coverslips were lifted off of the dish by adding PBS and gentle nudging with clean forceps. Coverslips were then flipped collagen-side up and allowed to sit in a culture hood until visibly dry. Next, another 0.75 mg/mL collagen solution was prepared and mixed with HL-60 cells to produce a solution of 5000-10,000 cells/µL. 30 µL of the cell-collagen suspension was pipetted onto the surface of a collagen-coated coverslip and immediately placed into a 37 °C incubator for 20 min in a covered petri dish.

Cell-laden gels were then fixed and immunostained, with all steps performed at room temperature. Here, a warmed solution containing 4% PFA, 5% sucrose, and PBS for 20 min. After fixation, coverslips were washed twice for 5 min with PBS. Cells were permeabilized with 0.5% Triton-X 100 in PBS for 10 min, washed twice for 5 min in PBS, and then incubated for 30 min with PBS and 0.05% Tween-20. Samples were then blocked using 20% goat serum in PBS with 0.05% Tween-20 for 30 min. Next, cells were immunolabeled with primary antibodies in PBS, 5% goat serum, and 0.05% Tween-20 for 1 hr. After incubation with primary antibody, samples were washed three times for 5 min with PBS and 0.05% Tween-20 and then stained with secondary antibodies, phalloidin, and DAPI, diluted in 1× PBS, 5% goat serum, and 0.05% Tween-20. Following fluorescent labeling, samples were washed three times for 5 min with PBS and 0.05% Tween-20 and once for 5 min in PBS. Samples were then stored at 4 °C in PBS or immediately imaged in PBS. Imaging was performed using 3D instantaneous structured illumination microscopy (iSIM) using a VisiTech iSIM mounted on a Nikon Ti Eclipse, with a ×100 1.35 NA silicone oil objective (Nikon). Images were captured using dual CMOS cameras (Hamamatsu, ORCA-fusion Gen III).

**Antibodies and dilution information.** Rabbit anti-human CORO1A (1:100, Cell Signaling, #D6K5B), Mouse anti-human FMNL1 (1:100, Santa Cruz Biotechnology, #sc-390466), Alexa Fluor 488 Phalloidin (1:400, Invitrogen, A12379), Goat anti-Rabbit IgG Alexa Fluor 594 (1:1000, Invitrogen, #A-11012), Rabbit anti-Mouse IgG Alexa Fluor 594 (1:1000, Invitrogen, #ab150116). Note: Control experiments using Alexa Fluor 488, 546, and 647 secondary antibodies without primary antibody each resulted in background signals that appeared as small, bright punctae within the cytoplasm of HL-60 cells and were not used for imaging.

### RNA-Seq
Total RNA was isolated $5 \times 10^6$ cells using the RNeasy Plus Mini kit (Qiagen, #74134). PolyA enrichment, RNA-seq, sequencing (Illumina), and data processing was performed by Novogene Corp. (Sacramento, CA). Sequencing reads were aligned to Homo Sapiens GRCh38/hg38 genomic research using Hisat2 (v2.0.5)[119]. Differential expression analysis of two conditions/groups, six biological replicates per condition, was performed using the DESeq2 package (v1.20.0)[120]. The statistical significance of differential gene expression was calculated in DESeq2 using a one-sided Wald test and adjusted for multiple comparisons using the Benjamini and Hochberg's approach. Genes with an adjusted $p$ value <0.05 were assigned as differentially expressed. Enrichment analysis was performed using clusterProfiler[121] to identify Gene Ontology (GO) and KEGG pathways with gene sets whose expression was significantly enriched by differential expressed genes. Gene set $p$ values estimate the statistical significance of gene set enrichment, calculated using a one-sided permutation test and adjusted for multiple comparisons using the Benjamini–Hochberg procedure.

## Reporting summary
Further information on research design is available in the Nature Portfolio Reporting Summary linked to this article.

## Data availability
Sequence data generated from CRISPRi screens and RNA-seq are uploaded to the Sequence Read Archive and can be accessed under BioProject accession code PRJNA976320. Genomic data for Homo Sapiens GRCh38/hg38 was used in the RNA-seq work. All data files used to generate figures are available in our GitHub repository, noted in the "Code availability" section below. Due to the large size of raw image files used for cell tracking, these are not included, but are available from the corresponding authors upon request. Source data are provided with this paper.

## Code availability
All processed data, code, and figure-generation scripts generated with the Python packages matplotlib[122] (v. 3.5.2) and seaborn[123] (v. 0.11.2) are publicly available as a GitHub repository [https://github.com/nbellive/CRISPRi_screen_HL60_pub][124].

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

## Acknowledgements

We thank members of the Theriot lab for useful discussions throughout this work. We also thank Dr. Alexander Leydon and Dr. Jennifer Nemhauser for access to their Bioruptor instrument, and Dr. Takato Imaizumi for access to Bio-Rad equipment used to perform Western blots. Research support was provided by the Howard Hughes Medical Institute (J.A.T.) and National Institutes of Health grants: DP2HD094656 (S.R.C.) and 5K99GM147355 (N.M.B.). S.R.C. is also supported by a Sidney Kimmel Foundation Kimmel Scholar Award, while N.M.B. was supported as a Fellow of the Jane Coffin Childs Memorial Fund for Medical Research. This article is subject to HHMI's Open Access to Publications policy.

HHMI lab heads have previously granted a nonexclusive CC BY 4.0 license to the public and a sublicensable license to HHMI in their research articles. Pursuant to those licenses, the author-accepted manuscript of this article can be made freely available under a CC BY 4.0 license immediately upon publication.

## Author contributions

Conceptualization: N.M.B. and J.A.T. Methodology: N.M.B. and M.F. Investigation: N.M.B., A.P.v.L., and E.A. Writing—original draft: N.M.B. and J.A.T. Writing—review and editing: M.F., A.P.v.L., E.A., and S.R.C. Visualization: N.M.B. Supervision: J.A.T. and S.R.C. Funding acquisition: J.A.T. and S.R.C.

## Competing interests

J.A.T. is chief scientific advisor at the Allen Institute for Cell Science (Seattle, WA, 98109). The authors otherwise declare no competing interests.
