## [Peer Review File · Nature Communications]

REVIEWER COMMENTS

Reviewer #1 (Remarks to the Author):

In this manuscript, Belliveau and colleagues present an interesting approach to screen for and analyze genes involved in various types of cell migration, including but not limited to 2D and amoeboid adhesion-independent migration, in the neutrophil-like HL60 cells. Rigorous data acquisition with proper controls and analysis makes the manuscript easy to understand. Most of the genes and pathways noted to be involved in HL60 cell migration and proliferation, validate previous findings. While it is difficult to digest the large data sets, the authors do a good job of highlighting key elements in the discussion and their data on mTOR-autophagy and integrin-recycling endosome pathways in the context of neutrophil proliferation and distinct modes of migration are of interest. Overall, the data obtained through the screen will be a great resource for the field of neutrophil differentiation and migration.

Specific comments:

1) While the HL60 cells were differentiated in RPMI media, what is the reason for changing the media to L-15 for the migration assays?

2) Why not use fMLF as a chemotactic source for the migration assays? This would add specificity to the signaling cascade and potentially give rise to more specific hits.

3) What is the temperature of low-melting agarose used to agarose overlay assays? There is a worry about cell viability.

4) PMID: 28916263, provides interestingly insight into the role of autophagy during neutrophil differentiation and should be discussed and cited.

5) X-axis labels on figure 3H are missing.

6) The authors mention that the FLCN and LAMTOR1 KD do not exhibit defects in the mTOR pathway in figure S3B. However, the presented data show decreased levels of S6K and Akt in the KD cells.

Quantification of several experiments would clarify this.

7) Unlike short-lived terminally differentiated neutrophils, macrophages are long lived. The data in figure 3E show a rescue of the cell density phenotype in the KD upon rapamycin treatment, which is an autophagy inducer, showing a role for autophagy during neutrophil differentiation as reported in PMID: 28916263. The authors mention that FLCN and LAMTOR1 KD cells have decreased expression of MPO and cathepsins but lack neutrophil elastase (crucial component of neutrophils). Coupled with the fact that these mutants have high CD11b levels and high adhesion, is it possible that the mutants are differentiating into macrophage-like cells, not neutrophil-like cells? This possibility should be explored.

8) The authors mention in figure 4C that the reason for the elongated morphology of ITGB2 knockdown cells is cell detachment. This needs to be experimentally validated as its not apparent from the data presented.

Reviewer #2 (Remarks to the Author):

The manuscript by Belliveau et al. describes and validates a CRISPR based screen to detect genes important for the proliferation, differentiation, and migration of human neutrophils. Overall, the paper is technically sound and represents a tour-de-force in terms of experimental rigor. We are particularly pleased and impressed with the robust experimental design that included high numbers of experimental replicates of each screen. This work provides detailed methods for useful assays to screen cell migration, and the inclusion of the full datasets in the supplement is an excellent resource for the field(s).

However, the paper in its current form is very difficult to read. We have outlined under "major points" a number of suggestions that the authors may wish to consider to increase the impact of their work:

Major points:

1. The title and abstract are misleading; a lot of the results are about cell growth and differentiation, and the observation that the authors see global similarity to differentiation in other systems. This

mismatch of expectations is unfortunate. The authors should adjust the title and abstract to match emphasis in differentiation in the results.

2. The authors should also consider providing background in the introduction about what pathways or genes are known to be specific to one, or more of these phenotypes. This would then make the genes described in the last paragraph of the results clear in how they help the reader appreciate the robustness of the data. (The authors may also wish to move those results to earlier in the paper.) Relatedly, the text states that " We also found a depletion of genes associated with granulopoiesis, suggesting that our screen data is identifying genes specific to neutrophil differentiation." without any indication of what genes one might expect to see in this list.

3. The manuscript is very long, and as written, it is hard to decipher what the take home message(s) are. It would improve both the readability and digestibility of the paper if the authors were to shorten the text. The authors may wish to: (1) identify a few main points and include concise summaries of each results section (these appear at the end of some but not all of the results sections); (2) reduce the number of very long (>50 words) sentences; (3) remove redundancy, particularly between results and discussion.

Minor points:

A. For each assay, what percent for cells were recovered from the starting population. Specifically, it seems possible that cells could get stuck in the channels during the transwell assay. Is there a way to estimate what fraction of cells might be excluded if they favor interacting with confined spaces, such as treating the membrane with Trypsin/EDTA and rinsing?

B. In Fig. 2B it is not obvious what cell migration assay was used. Were all three migration assays pooled? If so, how? This information should be outlined in the text and legend.

C. The chemotaxis and chemokinesis assays require the cells to move through a 3 micron pore. This step would require 3D motility, but this aspect is not discussed. It could be nice to link this idea to the FMNL1's importance in all motility screens, since it may help squeeze the nucleus.

D. On page 11, more of an explanation for the persistence measurements would be helpful. Specifically, if a random walk is 0, what do the negative values correspond to? On a related note, the diagram for persistence should be moved from Fig. 5B to Fig. 3G, where this concept is first shown.

E. The level of evidence provided for the claim about polarity in FLCN and LAMTOR1 knockdown cells (Page 11, Fig. 3F) should match the amount of evidence provided for the ITGB2 knockdown cells (Fig. 4C), which has a supplemental video and figure.

F. The axis labels on the bar graph are misaligned in Fig. 4B, resulting in what looks like mis-labeling of the graph contents.

G. Is there any evidence for the direction arrows shown in Fig 5 C-D, or is this inferred from the immunofluorescence? Either way, this should be explained in the figure legend. Some quantification of these phenotypes would also be helpful to provide context about how representative these phenotypes are.

Other points that the authors may wish to consider and/or address

i) Actin cartoons—subunits aren't staggered and the helicity is incorrect/inconsistent

ii) Fig 3B line to the right of the x-axis label

iii) The font sizes used in the figure are small, making them difficult to read. This is compounded in places by using black font over dark blue objects (see Fig. 2C for an example).

iv) Fig 3 E legend typo: "CIRSPRI"

v) Typo p 12 "migration of a stable knockdown line expression a sgRNA"

vi) The pale yellow used in Figure 7 is not detectable on some computer screens as it is so pale.

Reviewer #3 (Remarks to the Author):

Belliveau et al performed multiple pooled CRISPRi screens in an HL-60 model to understand proliferation, neutrophil differentiation, and cell migration assays. They identify factors in mTOR regulation and cell adhesion that affect these processes.

Major Comments:

1. More caveats need to be discussed in the interpretation and setup of the differentiation screen. There is a sentence about potential explanations for being a hit in the differentiation screen when the screen design is introduced, but more is needed. Couldn't some sgRNAs/genes that affect response to DMSO be miscategorized as hits in this screen? Wouldn't cells that just grow slowly look like differentiated cells leading to potential false negatives? It should be emphasized that other measurements of differentiation (markers/morphology) should be used to validate sgRNAs identified in the differentiation screen. The authors do these follow-ups, but it should be emphasized the need for these based on the screen design.

2. For validation experiments, were multiple sgRNAs tested? If not, were sgRNA off-targets addressed?

3. I found this sentence to be a strange claim. "One of the challenges with using fold-changes in sgRNA abundance to identify significant biological perturbations is that it provides little initial insight into the magnitude of biological effect." Isn't this exactly what fold changes are measuring? The data in 4D mostly counters this claim as well as there looks to be a correlation between Log2FC and the effect in the validation experiments. I think the authors' point is that you can't immediately convert the Log2FC into a value in the validation assay. However, as written, it sounds like Log2FCs aren't meaningful, which is not true.

Minor Comments:

4. What percent of DMSO-treated cells were CD11b positive in the screen? This would be helpful given these cells are taken into the cell-migration assays. Would non-differentiated HL-60s interfere with the migration assays? This could be more thoroughly explained.

5. Does differentiation change CRISPRi performance? Expression levels of Cas9 machinery may affect performance, which could change after differentiation. Given the screens yielded results, I think it still works, but was the CD4 test performed in differentiated cells?

6. In Figure 4, CRISPRi is misspelled in the captions.

Response to Reviewers for 'Cell migration CRISPRi screens in human neutrophils reveal regulators of context-dependent migration and differentiation state'

Nathan M. Belliveau, Matthew J. Footer, Emel Akdogan, Aaron P. van Loon, Sean R. Collins, and Julie A. Theriot

We thank the editor and expert reviewers for their thoughtful and constructive feedback on our manuscript. The text has been edited to read more concisely and improve clarity. Several additional experiments were also performed in relation to reviewer comments and we detail them in our point-by-point responses to the reviewer comments below.

Reviewer #1:

In this manuscript, Belliveau and colleagues present an interesting approach to screen for and analyze genes involved in various types of cell migration, including but not limited to 2D and amoeboid adhesion-independent migration, in the neutrophil-like HL60 cells. Rigorous data acquisition with proper controls and analysis makes the manuscript easy to understand. Most of the genes and pathways noted to be involved in HL60 cell migration and proliferation, validate previous findings. While it is difficult to digest the large data sets, the authors do a good job of highlighting key elements in the discussion and their data on mTOR-autophagy and integrin-recycling endosome pathways in the context of neutrophil proliferation and distinct modes of migration are of interest. Overall, the data obtained through the screen will be a great resource for the field of neutrophil differentiation and migration.

Specific comments:

1) While the HL60 cells were differentiated in RPMI media, what is the reason for changing the media to L-15 for the migration assays?

All cell culture growth and differentiation work was performed in a standard tissue culture incubator (5% CO₂ equilibration) in RPMI media. The change of media during migration assays was simply because our microscopy was performed without CO₂ equilibration. L-15 is buffered by phosphates and free base amino acids instead of sodium bicarbonate, providing a more suitably buffered environment during imaging. In our experience with these cells, we have not noticed a difference in migration behavior in these two media and all quantitative comparisons performed in the paper were made for migration in the same media.

We have made an additional note in the Methods section entitled, '*Cell migration assays using individual sgRNA CRISPRi lines*,' where L-15 is first noted: '*We note that the switch from RPMI to L-15 was made to avoid the need for CO₂ in our microscope system for pH buffering.*'

2) Why not use fMLF as a chemotactic source for the migration assays? This would add specificity to the signaling cascade and potentially give rise to more specific hits.

Thank you for the suggestion. We did consider this early on and performed preliminary migration experiments using track-etch membranes with a fMLF chemotactic source. However, we found that serum provided a robust and reliable migration response and felt that it would provide a more useful starting point to assess the global aspects of cell migration rather than the signaling pathways specific to one receptor. Indeed, in future work we will be interested in applying fMLF (and other alternative chemotactic sources) to allow us to consider specificity. Given that fMLF will still stimulate migration in both chemokinesis and chemotaxis formats ¹, it remains an interesting and open question how such a screen might compare with the results of our serum stimulated cells.

In the *Discussion* section we now write,

Chemotaxis, at least in the context of serum stimulation as explored here, likely acts to more efficiently guide the movement of cells through the pores of the track-etch membrane, but our results indicate that the underlying molecular mechanisms of directed and spontaneous cell motility are essentially identical. For more specific chemoattractants like fMLF, we still expect an increase in migratory activity during both chemokinesis and chemotaxis ¹, and we hypothesize that the similarity between chemotaxis and chemokinesis may be more broadly applicable.

3) What is the temperature of low-melting agarose used to agarose overlay assays? There is a worry about cell viability.

This is a valid concern. However, we use Invitrogen UltraPure™ Low Melting Point Agarose (cat# 16520050, ThermoFisher), which remains fluid at 37°C and will set at room temperature (below 25°C). To generate the agarose overlay, we initially prepare a 2x concentrated agarose liquid solution, allow it to cool down sufficiently and dilute it 1:1 with pre-heated 37°C L-15 media.

We have edited the relevant *Methods* section entitled, '2D migration using agarose overlay and fMLF photo-uncaging', to more clearly state this,

Cells were allowed to adhere to the glass for 5 min, before a 195 µl layer of 1.5% low-melt agarose solution in L-15 (Goldbio, #A-204-25) was mixed 1:1 with L-15 + 10% hiFBS media (warmed to 37°C) and overlaid on top. We note that the initial agarose mixture was allowed to cool to approximately 37°C prior to preparing the final dilution and then used immediately.

4) PMID: 28916263, provides interesting insight into the role of autophagy during neutrophil differentiation and should be discussed and cited.

We thank you for the reference and have now cited the paper. We agree that this paper provides relevant results to the work presented here. We believe it provides valuable support for our work on mTORC1 signaling (e.g. FLCN and LAMTOR1 knockdown lines). Our RNA-seq data identifies altered downstream targets of mTORC1 signaling that suggests an inhibitory effect on autophagy. In addition, our observation of changes in granule gene expression also appear consistent with the observations made in this reference when autophagy was directly inhibited (i.e. an increase in *mpo* and decrease in *mmp9*). One notable difference in our FLCN and LAMTOR1 knockdown dHL-60 cells is that we observe normal to elevated levels of CD11b, while they observe defective induction of

CD11b following inhibition of autophagy during differentiation.

With the work from Bhattacharya et al.² (PMID: 26344765), which looked at functional changes in neutrophils following an inhibition of autophagy, at this point it remains difficult to rule out whether the genetic disruptions are specifically affecting differentiation, function, or a combination of the two. Furthermore, as brought up in comment 7 below, another intriguing possibility is that altered mTORC1 signaling may be playing a more regulatory role in differentiation and skewing differentiation toward a more macrophage-like character.

To highlight these various points, in the the *Results* section titled, '*Disruption of folliculin and Ragulator-Rag signaling pathways results in altered but active mTORC1 signaling in dHL-60 neutrophils,*' we now include the following text,

Direct inhibition of autophagy has been shown to affect both neutrophil differentiation and effector function^{2,3} and our data suggest that FLCN and LAMTOR1 knockdown play a similar inhibitory role through altered mTORC1 activity. These knockdown lines also showed changes in the expression of genes associated with neutrophil degranulation, including an increase in mpo and a decrease in mmp9 (Supplementary Fig. 3d, Supplementary Data 6 and 7). This is also observed following the inhibition of autophagy during differentiation and may be a reflection of incomplete differentiation³.

Supplementary Data 6 and 7 are now included and contain the associated differential RNA-seq expression analysis for the differentiated FLCN and LAMTOR1 knockdown lines.

5) X-axis labels on figure 3H are missing.

Thank you for catching this error. We have made the correction.

6) The authors mention that the FLCN and LAMTOR1 KD do not exhibit defects in the mTOR pathway in figure S3B. However, the presented data show decreased levels of S6K and Akt in the KD cells. Quantification of several experiments would clarify this.

We have quantified Western blot data from several rounds of experiments and now include that data in place of the total protein loading images that were previously included as Supplementary Fig. 3C (shown below, Fig. R1, with the blots provided in the zipped source data file). Here, band intensities were normalized to a loading control (Gapdh) and then relative intensities were calculated in order to compare multiple blots. Overall we find no significant difference in the LAMTOR1 and FLCN knockdown lines relative to our control line. While the data may actually suggest an increase in mTOR phosphorylation, the effect size is within the noise of our quantitation.

We also note that there may have been some confusion regarding the specific bands that correspond to S6K pThr389. Specifically, our antibody labels two protein isoforms (p85 and p70) and we have now identified these in Supplementary Figure 3B. A key difference between these protein isoforms is the addition of a nuclear localization sequence in the N-terminus of p85. However, current literature suggests it is still present in the cytosol⁴ and we have chosen to include intensities from both proteins in the quantitation performed. For Akt, the slightly lower intensity for FLCN and LAMTOR1 knockdown lines appear to be in part due to lower sample loading.

Note that to better emphasize that mTOR kinase activity remains active following knockdown of LAMTOR1 and FLCN, which is perhaps the most important conclusion of these experiments, we have edited the main text in Section ‘*Disruption of folliculin and Ragulator-Rag signaling pathways results in altered but active mTORC1 signaling in dHL-60 neutrophils*’, to state that ‘*Western blot analysis ... showed that kinase activity of mTORC1 and mTORC2 remained active following knockdown of LAMTOR1 and FLCN.*’

Figure R1: Quantification of western blot data associated with mTORC1 and mTORC2 signaling. Western blot band intensities were normalized to total protein loading, estimated from an Alexa Fluor 790 GAPDH antibody. Relative normalized intensity values are compared to the sgControl cell line. Bar plot shows the average value and standard deviation, while individual points represent individual replicates blots (mTOR, n = 3; mTOR pSer2448, n = 4; S6K pThr389, n = 3; Akt pSer473, n = 2). Quantitation of S6K protein included band intensities for both p85 and p70 isoforms.

7) Unlike short-lived terminally differentiated neutrophils, macrophages are long lived. The data in figure 3E show a rescue of the cell density phenotype in the KD upon rapamycin treatment, which is an autophagy inducer, showing a role for autophagy during neutrophil differentiation as reported in PMID: 28916263. The authors mention that FLCN and LAMTOR1 KD cells have decreased expression of MPO and cathepsins but lack neutrophil elastase (crucial component of neutrophils). Coupled with the fact that these mutants have high CD11b levels and high adhesion, is it possible that the mutants are differentiating into macrophage-like cells, not neutrophil-like cells? This possibility should be explored.

This is an interesting possibility and one that has also come up during our own group meetings. The uHL-60 cells can be made to differentiate into macrophage-like cells by incubating them with phorbol myristate acetate (PMA).

We performed some initial exploration into this by first identifying genes that are expected to exhibit differential expression between neutrophil and macrophage cell types. We then asked whether the RNA-seq expression changes in our neutrophil-differentiated FLCN and LAMTOR1 knockdown lines (relative to our sgControl line) were consistent with a shift toward a macrophage-like cell. We provide those details below and note that this idea does indeed appear to have some support. A more substantial undertaking to fully clarify this will be needed, but we believe these findings warrant inclusion of this idea and associated analysis. We do so in the *Results* section titled, ‘*Disruption of*

folliculin and Ragulator-Rag signaling pathways results in altered but active mTORC1 signaling in dHL-60 neutrophils' with the additional text,

In our differentiated FLCN and LAMTOR knockdown lines, we also observed an increase in expression of several genes associated with macrophage differentiation, including ccr5, cd163, cd64, cd71. uHL-60 cells are multipotent cells that can be differentiated into other cell types, including macrophages⁵. Given the observed induction of CD11b and fMLF receptor in dHL-60 FLCN and LAMTOR1 knockdown lines (Fig. 4c, Supplementary Fig. 3a) and that these cells are longer lived, we reasoned that FLCN and LAMTOR1 knockdown might be skewing their differentiation trajectory away from a purely neutrophil-like character and towards a more macrophage-like state. To test this possibility, we mined an available RNA-seq dataset that measured gene expression changes during differentiation of uHL-60 cells into both neutrophil-like cells and macrophage-like cells⁶, we identified a variety of genes including cd52 that show higher differential expression in macrophage-like cells (Supplementary Table 1). Using flow cytometry, we found that both FLCN and LAMTOR1 knockdown cell lines exhibited higher expression of CD52 than control lines after DMSO-triggered differentiation, consistent with this hypothesis (Supplementary Figure 4). While further work is needed to fully dissect the possibility that these knockdowns alter the trajectory of cell fate, it does help to explain the observed pattern of enrichment of dHL-60 cells in our differentiation screen following knockdown of FLCN and LAMTOR1.

Additional analysis:

We used data from Ramirez et al. 2017, *Cell Systems* (PMID: 28365152) to identify genes that exhibited differential expression between neutrophil and macrophage differentiation. In that work, the authors performed RNA-seq during differentiation of uHL-60 cells into neutrophil-like cells (induced with all-trans retinoic acid, instead of DMSO as performed here) and macrophage-like cells (induced using PMA). From their analysis they generated 13 clusters of genes that showed different trends in differential expression during differentiation.

In Table R1 we show a set of genes expected to show increased expression during macrophage-like differentiation, but decrease during neutrophil differentiation. For both dHL-60 FLCN and LAMTOR1 knockdown lines, all genes show an increase in expression relative to the control cell line. This is the expected position shift if cells were becoming more macrophage-like.

Gene	Log2 fold change (day5+DMSO, sgFLCN vs sgControl)	Log2 fold change (day5+DMSO, sgLAMTOR1 vs sgControl)
CSF1	1.1	1.62
CD52	3.23	3.77
EMP1	2.4	3.62
CD9	1.36	3.16
FCRLA	4.82	5.52
FCRLB	0.78	1.52
ACTN1	0.30	0.45
MYO1C	0.47	0.47

Table R1: Genes expected to show increased expression following macrophage-like differentiation, but decrease during neutrophil differentiation. Log₂ fold-change values in RNA transcript expression are shown for differentiated FLCN and LAMTOR1 knockdown lines relative to control cells. Genes were identified from the data in Ramirez et al. 2017, *Cell Systems* (PMID: 28365152).

Similarly, in Table R2 we show genes where expression is expected to decrease during macrophage-like differentiation, but increase during neutrophil differentiation. Here we see a predominantly negative shift in line with the shift toward macrophage cells.

Gene	Log2 fold change (day5+DMSO, sgFLCN vs sgControl)	Log2 fold change (day5+DMSO, sgLAMTOR1 vs sgControl)
S100A8	-1.18	-2.40
PKN1	-0.58	-0.77
MYADM	-0.34	not present in data
SLC9A3R1	-0.60	-0.86
TESC	0.24	-0.36
CSF3R	-0.72	-1.47
CD4	-0.23	-0.53
FGR	-0.25	-0.20
LRG1	-0.67	-0.82
LSP1	0.15	0.15

Table R2: Genes expected to show increased expression following macrophage-like differentiation, but decrease during neutrophil differentiation. Log₂ fold-change values in RNA transcript expression are shown for differentiated FLCN and LAMTOR1 knockdown lines relative to control cells. Genes were identified from the data in Ramirez et al. 2017, *Cell Systems* (PMID: 28365152).

Importantly, we also see a positive shift in expression of several common macrophage markers, as shown in Table R3.

Gene	Log2 fold change (day5+DMSO, sgFLCN vs sgControl)	Log2 fold change (day5+DMSO, sgLAMTOR1 vs sgControl)
CD163	3.62	3.52
CCR5	2.93	4.04
CD14	-0.67	-1.41
CD16 (FCGR3A)	not present in data	-2.63
CD64 (FCGR1)	0.86	0.42
CD71 (TFRC)	2.17	2.0
CD68	not present in data	not present in data

Table R3: Differential gene expression of common macrophage markers. Log₂ fold-change values in RNA transcript expression are shown for differentiated FLCN and LAMTOR1 knockdown lines relative to control cells. Genes were identified in the Abcam antibody panel cat. #ab254013 and the Bio-Rad resource, <https://www.bio-rad-antibodies.com/macrophage-m1-m2-tam-tcr-cd169-cd-markers-antibodies.htm>.

Lastly, we selected one of the genes, CD52, and performed flow cytometry immunofluorescence to confirm its expression change at the protein level. As can be seen in Fig. R2A, control cells show a slight decrease in CD52 expression following neutrophil differentiation (top row, shift from gold to blue), but a large increase in expression upon macrophage differentiation. This was the expected relative changes based on the data from Ramirez et al. When performing neutrophil differentiation of the FLCN and LAMTOR knockdown lines, however, we observe a mean positive shift in CD52 expression (Fig. R2A, bottom two rows). While clearly not as dramatic a shift in expression as the observed during PMA-induced macrophage-differentiation, this data is consistent with a shift in differentiation trajectory.

Figure R2: Surface expression of CD52 in uHL-60 cells, dHL-60 neutrophil-like, and dHL-60 macrophage-like cells following knockdown of FLCN and LAMTOR1.

(A) Surface expression of CD52 was measured by immunofluorescence flow cytometry (see *Methods* section of main text for additional details). Neutrophil-like cells were prepared by inducing differentiation with DMSO as described in the main text. Macrophage-like cells were generated by incubating cells with PMA at a concentration of 50 ng/mL for 5 days. For both differentiation conditions, media was replaced with fresh media, including the differentiation reagent, at day 3 following the initiation of differentiation. Black arrows qualitatively indicate the sign shift in expression following DMSO mediated differentiation.

(B) Immunofluorescence flow cytometry of undifferentiated uHL-60 cells, dHL-60 neutrophil-like cells, and dHL-60 macrophage-like cells using an isotype control. This plot highlights that some of the observed change in signal intensity is likely due to differences in autofluorescence or antibody binding across the different cell types.

8) The authors mention in figure 4C that the reason for the elongated morphology of ITGB2 knockdown cells is cell detachment. This needs to be experimentally validated as its not apparent from the data presented.

We thank the reviewer for pointing this out. We noticed that we did not reference the supplemental video, which more clearly shows the ITGB2 knockdown cell appearing poorly attached during much of the video. We apologize for this and have updated the text. We have also selected a different frame for the figure panel from this video that more clearly distinguishes the control and ITGB2

knockdown lines (Fig. 5b).

We have also performed an adhesion assay to more quantitatively show the poorer adhesion in the ITGB2 knockdown cell line. Here, 400,000 cells were suspended in 250 μ L RPMI supplemented with 10% heat-inactivated FBS and placed in wells of a 24-well TC-treated multiwell plate. After incubating cells for 45 minutes, the media and non-adhered cells were aspirated. Following a gentle wash where 500 μ L culture media was added and then aspirated, the remaining cells were resuspended and counted. Figure R2 shows the results for our control and ITGB2 knockdown lines, where we plot adhered cell fraction by normalizing cell counts relative to wells where no cells were aspirated. This data is included as Supplementary Fig. 5c.

Figure R2: Quantification of adhesion phenotype in ITGB2 knockdown cell line. Error bars indicate standard deviation across individual wells. Data points correspond to measurements from 6 individual wells (assay details are also included in the *Methods* section of main text).

Reviewer #2:

The manuscript by Belliveau et al. describes and validates a CRISPR based screen to detect genes important for the proliferation, differentiation, and migration of human neutrophils. Overall, the paper is technically sound and represents a tour-de-force in terms of experimental rigor. We are particularly pleased and impressed with the robust experimental design that included high numbers of experimental replicates of each screen. This work provides detailed methods for useful assays to screen cell migration, and the inclusion of the full datasets in the supplement is an excellent resource for the field(s).

However, the paper in its current form is very difficult to read. We have outlined under “major points” a number of suggestions that the authors may wish to consider to increase the impact of their work:

Major points:

1. The title and abstract are misleading; a lot of the results are about cell growth and differentiation, and the observation that the authors see global similarity to differentiation in other systems. This mismatch of expectations is unfortunate. The authors should adjust the title and abstract to match emphasis in differentiation in the results.

We appreciate the suggestion to make the title and abstract more precise and believe our updated versions now better encompass the significant results that were provided by our work on growth and differentiation.

We have removed ‘Cell migration CRISPRi screens’ from the title to remove the sole emphasis on those screens. The updated title is ‘*Whole-genome screens reveal regulators of differentiation state and context-dependent migration in human neutrophils*’.

For the abstract, we believe we have found a better balance in our summary of the work and have modified the text as follows,

Neutrophils are the most abundant leukocyte in humans and provide a critical early line of defense as part of our innate immune system. We have performed a comprehensive, genome-wide assessment of the molecular factors critical to proliferation, differentiation, and cell migration in a neutrophil-like cell line. Multiple migration screen strategies were developed to specifically probe directed (chemotaxis), undirected (chemokinesis), and 3D amoeboid cell migration in these fast-moving cells. We identify a role for mTORC1 signaling in cell differentiation, which influences neutrophil abundance, survival, and migratory behavior. Across our individual migration screens, we identify genes involved in adhesion-dependent and adhesion-independent cell migration, protein trafficking, and regulation of the actomyosin cytoskeleton. This genome-wide screening strategy, therefore, provides an invaluable approach to the study of neutrophils and provides a resource that will inform future studies of cell migration in these and other rapidly migrating cells.

2. The authors should also consider providing background in the introduction about what pathways or genes are known to be specific to one, or more of these phenotypes. This would then make the genes described in the last paragraph of the results clear in how they help the reader appreciate the robustness of the data. (The authors may also wish to move those results to earlier in the paper.)

Relatedly, the text states that “ We also found a depletion of genes associated with granulopoiesis, suggesting that our screen data is identifying genes specific to neutrophil differentiation.” without any indication of what genes one might expect to see in this list.

In order to address the concerns related to text length and readability, we have chosen not to go into additional detail about what pathways or genes are already known to play a role in and across the various phenotypes. However, we have edited the introductory text to reference recent efforts that highlight neutrophil differentiation and functionality is more complex than previously thought and is actively being investigated.

On the specific sentence related to granulopoiesis, we have updated the text to now indicate specific genes. Specifically, we now write,

‘We also found a depletion of key regulatory genes associated with granulopoiesis including the transcription factors Fli-1 (FLI1) and the CCAAT/enhancer binding proteins, C/EBP α (CEBPA) and C/EBP β (CEBPE), suggesting that our screen data is identifying genes specific to neutrophil differentiation.’

3. The manuscript is very long, and as written, it is hard to decipher what the take home message(s) are. It would improve both the readability and digestibility of the paper if the authors were to shorten the text. The authors may wish to: (1) identify a few main points and include concise summaries of each results section (these appear at the end of some but not all of the results sections); (2) reduce the number of very long (>50 words) sentences; (3) remove redundancy, particularly between results and discussion.

We appreciated this concern and believe we have made major strides to improve readability. We describe key edits below.

- The main text (Introduction, Results, and Discussion) have been edited substantially, with about 1,500 words removed.
- While the main results remain largely intact, we have removed some portions of the text exploring genes identified in the screens that were not followed up with additional experimental work. Specifically, within the sections on differentiation, we removed text associated with SUMOylation genes and some of the text on regulatory proteins have been removed.
- The text from the final section ‘CRISPRi screens of cell migration provide a rich resource for studying rapidly migrating cells’ has been made more concise and moved a bit earlier in the text, along with Figure 7.
- We have also removed the section, ‘*Knockdown of folliculin and LAMTOR1 increase substrate adhesion but reduce migratory persistence and chemotaxis sensitivity*,’ where we characterized migration of the LAMTOR1 and FLCN knockdown lines. This analysis has been integrated into a section called ‘*Migration through track-etch membranes is dominated by genes associated with cell adhesion*’.
- The main text figures have been split up differently, based on the edits, and are now spread across 8 main text figures.
- We have added concise summaries for each of the *Results* sections.
- We have removed redundancy between the *Results* and *Discussion* sections and use the discussion to predominantly focus on the broader take-aways from the work.

Minor points:

A. For each assay, what percent for cells were recovered from the starting population. Specifically, it seems possible that cells could get stuck in the channels during the transwell assay. Is there a way to estimate what fraction of cells might be excluded if they favor interacting with confined spaces, such as treating the membrane with Trypsin/EDTA and rinsing?

Given the rapid migration rate of these cells and the relatively short path length of the track-etch membrane pores (~20 μm , or about 1-2 cell lengths), over the time scale of these experiments (2 hr and 6 hr time points) we do not expect a substantial population of cells to remain in the pores as a consequence of favoring confined spaces. Given the still relatively low adherence of this cell type, it appears that upon exiting the pores the cells fall off the membrane and end up in the bottom of the well. Because of this, cells will be less likely to equilibrate between favoring a confined vs. unconfined environment. Regarding the specific numbers of cells recovered, we generally expect a loss on the order of 10% of the starting population. However, this likely includes cell losses due to residual cells remaining in the top/bottom reservoirs during collection and from liquid handling.

Your point about favoring confined spaces is very interesting and fun to consider, but we believe this would require a different selection strategy to properly assay such a phenotype. Specifically, if we could provide cells with a confined vs. unconfined environment and allow them to equilibrate between the two, would it be possible to identify gene perturbations that alter their spatial preference? We could envision an alternative selection assay, where cells are seeded on a surface containing small pores (e.g. using a microfabricated surface with PDMS patterning). After some time, there may be a preference to reside in or out of the pore and this preference could be perturbed through genetic perturbations.

B. In Fig. 2B it is not obvious what cell migration assay was used. Were all three migration assays pooled? If so, how? This information should be outlined in the text and legend.

The cell migration data in Fig. 2B was indeed pooled and we have now updated the text and legend to clarify this. Specifically, the data points represent normalized \log_2 fold-change, calculated by averaging across all individual migration screen replicates. We have now updated the text and caption legend for Fig. 2B to specifically state this (now Fig. 2b).

C. The chemotaxis and chemokinesis assays require the cells to move through a 3 micron pore. This step would require 3D motility, but this aspect is not discussed. It could be nice to link this idea to the FMNL1's importance in all motility screens, since it may help squeeze the nucleus.

This is a reasonable idea and we appreciate the suggestion. In the results section entitled, '*CRISPRi screen identifies genes important for 3D amoeboid cell migration*' we now write,

We used immunofluorescence to determine the localization of these proteins in wild-type cells. In contrast to the expected leading edge localization of well-characterized formins like mDia1/2⁷, FMNL1 was rear-localized and often directly behind the nucleus (Fig. 6c, Supplementary Fig. 6a). This is consistent with recent work in T-cells⁸, who found similar localization and hypothesized that formin-like 1 may support actin polymerization to aid in squeezing the nucleus through tight endothelial barriers. Indeed, knockdown of FMNL1 was also identified in our track-etch membrane-

based screens and may more specifically be supporting movement as cells squeeze through small pores.

D. On page 11, more of an explanation for the persistence measurements would be helpful. Specifically, if a random walk is 0, what do the negative values correspond to? On a related note, the diagram for persistence should be moved from Fig. 5B to Fig. 3G, where this concept is first shown.

We have moved the diagram where the concept is first shown (now, Fig. 5d) and expanded on our explanation of the persistence measure. Specifically, we have updated that text as follows,

We employed a Bayesian inference algorithm based on a model for a heterogeneous random walk⁹ to calculate a migratory persistence metric for each cell. In this model, a persistence value of zero corresponds to a non-persistent, diffusive movement. Persistence values closer to -1 indicate anti-persistent, reversive movement, while values closer to +1 indicate more persistent migration.

E. The level of evidence provided for the claim about polarity in FLCN and LAMTOR1 knockdown cells (Page 11, Fig. 3F) should match the amount of evidence provided for the ITGB2 knockdown cells (Fig. 4C), which has a supplemental video and figure.

We agree with this and have generated an equivalent set of supplemental data for our analysis of FLCN and LAMTOR1 knockdown cell lines. In Supplementary Fig. 5b we have included a set of stills for the FLCN and LAMTOR1 knockdown lines and a video in Supplementary Video 2.

F. The axis labels on the bar graph are misaligned in Fig. 4B, resulting in what looks like mis-labeling of the graph contents.

Thank you for catching this. We have corrected the alignment of the text (now Fig. 5a).

G. Is there any evidence for the direction arrows shown in Fig 5 C-D, or is this inferred from the immunofluorescence? Either way, this should be explained in the figure legend. Some quantification of these phenotypes would also be helpful to provide context about how representative these phenotypes are.

Thank you for the concern. The direction arrows are only qualitative since the cells were fixed prior to immunofluorescence, but are based on our experience imaging live cells containing GFP-tagged β -actin¹⁰. We now note that the arrows are qualitative in the caption legend.

As suggested, we have also performed additional quantification to provide a better sense of how representative the selected images are for localization in migrating cells. We show that analysis below in Fig. R3 and have also included it as Supplementary Fig. 6.

Figure R3: Quantification of formin-like protein 1 and coronin 1A during migration of dHL-60 neutrophils in collagen.

(A) Fluorescent histograms of immunofluorescence images for formin-like protein 1 (top) and DAPI DNA stain (bottom) across 7 cells. The dark line represents an averaged histogram across all cell data. Schematic insets identify the localization of fluorescent signal intensity.

(B) Fluorescent histograms of immunofluorescence images for coronin 1a (top) and DAPI DNA stain (bottom) across 8 cells. The dark line represents an averaged histogram across all cell data. Schematic insets identify the localization of fluorescent signal intensity.

For (A) and (B), Image data represent cells that were clearly migrating based on their polarization and generally more intense localization of F-actin at the cell front. In order to generate histograms, images were rotated to orient the cell horizontally, approximately as shown in the schematic illustration.

Other points that the authors may wish to consider and/or address

i) Actin cartoons—subunits aren't staggered and the helicity is incorrect/inconsistent

Thank you for identifying this. We have edited the schematics that were present in Figure 4 and Figure 7 (now Figure 5 and Figure 7, respectively). We have also flipped the schematic of Figure 4B 180 degrees so that the extracellular integrin domains are pointing down, more consistent with Figure 7.

ii) Fig 3B line to the right of the x-axis label

This line was erroneously added and has been removed. We have also edited the Figure 3 axis labels for FPR1 to be consistent with the ITGAM labels (i.e. 'fMLF receptor 1 (FPR1) expression [a.u.]' instead of 'FPR1 expression [a.u.]').

iii) The font sizes used in the figure are small, making them difficult to read. This is compounded in places by using black font over dark blue objects (see Fig. 2C for an example).

Thank you for noting this. We have made an effort to increase font size throughout the figures and for Fig. 2C in particular (now Fig. 2d), we have adjusted the text colors to improve clarity.

iv) Fig 3 E legend typo: "CIRSPRi"

Thank you, this has been corrected.

v) Typo p 12 "migration of a stable knockdown line expression a sgRNA"

Thank you for catching this. We have edited the text to say '*migration of a stable knockdown line expressing a sgRNA*'.

vi) The pale yellow used in Figure 7 is not detectable on some computer screens as it is so pale.

Thank you, we have changed the pale yellow to a darker shade.

Reviewer #3:

Belliveau et al performed multiple pooled CRISPRi screens in an HL-60 model to understand proliferation, neutrophil differentiation, and cell migration assays. They identify factors in mTOR regulation and cell adhesion that affect these processes.

Major Comments:

1. More caveats need to be discussed in the interpretation and setup of the differentiation screen. There is a sentence about potential explanations for being a hit in the differentiation screen when the screen design is introduced, but more is needed. Couldn't some sgRNAs/genes that affect response to DMSO be miscategorized as hits in this screen? Wouldn't cells that just grow slowly look like differentiated cells leading to potential false negatives? It should be emphasized that other measurements of differentiation (markers/morphology) should be used to validate sgRNAs identified in the differentiation screen. The authors do these follow-ups, but it should be emphasized the need for these based on the screen design.

We appreciate the concern here and the need for further clarity over what our differentiation screen has assessed. As with most drop-out style pooled screens, there may be gene perturbations that lead to a change in log₂ fold-change, but are less specific to the biological function under study.

One other approach that we envisaged in performing a genome-wide screen of differentiation was to sort our pooled CRISPRi dHL-60 library based on the level of induction of a surface marker like CD11b. However, it is worth noting that while this may have identified regulatory genes like SPI1 that exhibit reduced CD11b induction, it would likely have missed other relevant genes. For example, the genes identified along the mTORC1 pathway exhibited only slightly elevated levels of CD11b but clearly play a role in the differentiation state and migratory behavior of these cells.

We have edited the text to highlight the specific expectations from our approach and point out potential caveats.

In the first results section we now write,

To derive dHL-60 cells, we induced differentiation by incubating uHL-60 cells with 1.57% DMSO for five days, which provides a near-complete differentiation of the cell population into CD11b+ neutrophil-like cells¹². Viable dHL-60 cells were isolated using density gradient centrifugation to remove dead cells and cellular debris following our differentiation protocol. We compared sgRNA abundance between dHL-60 and uHL-60 cells to identify gene perturbations that altered the abundance of cells during differentiation and, therefore, could serve as indicators of altered differentiation. Here, we identified 989 genes that were depleted and 869 genes that were enriched relative to our control sgRNAs (Fig. 1e). The ratio of enriched sgRNAs to depleted sgRNAs was strikingly different from our proliferation screen, where only ~2% of knockdowns led to an enrichment of sgRNAs.

In addition, the Discussion now includes the following text,,

With roughly 10¹¹ neutrophils produced in the bone marrow each day¹⁴, any process that impacts neutrophil abundance will severely influence their protective capabilities. As such, we chose to

identify genes relevant to differentiation by looking at changes in cell abundance between differentiated and undifferentiated cells following gene knockdown. While some of the results from our screen of differentiation may reflect specific genetic sensitivities of the HL-60 leukemia cell line and our differentiation protocol, we were encouraged by the identification of key transcriptional regulatory genes, including CEBPA, CEBPE, and SPI1, which are known to be involved in neutrophil differentiation. Beyond these genes, we found that perturbations along the FLCN-RagC/D signaling axis of mTORC1 signaling impacts differentiation and survival, and also dramatically altered cells' migratory phenotypes. We hypothesize that these effects proceed via a non-canonical mechanism of mTORC1 regulation that has only recently begun to be elucidated^{15,16}.

In the original text we included a comment on the potential role of DMSO and have kept this in the Discussion,

While the mechanism behind DMSO-mediated differentiation of myeloid precursors remains poorly understood¹⁷, DMSO can alter membrane permeability and impair mitochondrial function^{18,19} and may act as a chemical insult that drives cell differentiation by impairing cellular homeostasis. Mitochondria, in particular, are hubs of metabolic signaling that produce molecules that modulate cellular function, gene expression, and can alter differentiation state^{20,21}.

2. For validation experiments, were multiple sgRNAs tested? If not, were sgRNA off-targets addressed?

For validation experiments, only the sgRNA with the most significant effect was selected for follow up work. We cannot completely rule out the possibility of off-target effects. However, we did confirm sgRNA knockdown specifically for most of the cell lines where follow up work was performed.

- ITGB2: We confirmed loss of extracellular protein by flow cytometry immunofluorescence (Fig. R4A).
- FLCN, LAMTOR1, SPI1, and ATIC knockdown lines: RNA-seq analysis showed that these genes were among the most differentially expressed genes relative to our cell line containing a control sgRNA (Fig. R4B).
- FMNL1: We confirmed loss of protein by immunofluorescence (Fig. R4C).
- We also note that we measured ITGA1 by flow cytometry immunofluorescence, but fluorescence intensity values from our ITGA1 antibody probe were quite low relative to our isotype control, making it difficult to quantify knockdown. Encouragingly, the fluorescence histogram for ITGA1 in our ITGA1 knockdown line appeared almost indistinguishable from our isotype control, suggesting there is knockdown of this protein.

In section, '*CRISPRi cell line construction*' of *Methods* we now note: *Knockdown was confirmed in the ITGB2 knockdown line by flow cytometry immunofluorescence, in the FLCN, LAMTOR1, SPI1, and ATIC knockdown lines by RNA-seq, and in the FMNL1 knockdown line by immunofluorescence microscopy.*

While the estimated knockdown in our SPI1 knockdown line was not as dramatic as we might have expected, we do observe differential expression of expected genes relative to our control sgRNA cell line in dHL-60 cells. This includes SPI1 regulated targets (e.g. *egr1*, *egr3*, *nfe2* identified in ref.⁶) and we also observe a lack in the expected induction of CD14, CD18, and CD11b transcripts¹¹.

Lastly, we emphasize that the screens employed three unique sgRNA per gene and we identified gene hits based on the averaged log₂ fold-change values across all three sgRNA. This should

alleviate some concern over potential off-target specificity since more than one sgRNA generally led to a significant effect.

Figure R4: Validation of gene knockdown in ITGB2, FLCN, LAMTOR1, SPI1, ATIC, and FMNL1 knockdown lines.

- (A) Integrin β_2 knockdown was estimated at the protein level using immunofluorescence flow cytometry data (related to data in histograms Figure 8B of main text). Measurements reflect the average fluorescence intensity from three sample measurements performed over two days. Intensity values have been background subtracted using the fluorescence signal from isotype antibody controls.
- (B) RNA-seq data was used to estimate gene knockdown at the transcript level in FLCN, LAMTOR1, SPI1, and ATIC knockdown lines. Following count normalization of the RNA-seq data to allow for differential expression comparisons across samples (DESeq2's median of ratios method¹³), the change in mRNA expression was calculated relative to our control cell line. This was performed for each sample and time point where data was available (related to Fig. 4d of main text). Error bars represent standard deviation across 6 independent RNA-seq samples.
- (C) Microscopy based immunofluorescence was used to estimate knockdown in our FMNL1 knockdown line. Using data similar to that shown in Fig. 6c, we estimated the average fluorescence intensity in individual cells in a FMNL1 knockdown line ($n = 34$ cells) and control cell line ($n = 27$ cells). Error bars represent standard deviation in average intensity values, which were also background subtracted to remove autofluorescence.

3. I found this sentence to be a strange claim. "One of the challenges with using fold-changes in sgRNA abundance to identify significant biological perturbations is that it provides little initial insight into the magnitude of biological effect." Isn't this exactly what fold changes are measuring? The data in 4D mostly counters this claim as well as there looks to be a correlation between Log2FC and the effect in the validation experiments. I think the authors' point is that you can't immediately convert the Log2FC into a value in the validation assay. However, as written, it sounds like Log2FCs aren't meaningful, which is not true.

We thank you for the note. The reviewer is correct in their interpretation that we wanted to be able to relate log₂ fold-change values to an absolute measure, such as the specific fraction of cells migrated.

We have edited the text to now simply state,

In order to confirm the measured \log_2 fold-change values for cells reaching the lower reservoir in the large-scale screen with a more direct measurement of transmigration, we assayed the fraction of cells that migrate across the track-etch membrane in individual knockdown cell lines.

Minor Comments:

4. What percent of DMSO-treated cells were CD11b positive in the screen? This would be helpful given these cells are taken into the cell-migration assays. Would non-differentiated HL-60s interfere with the migration assays? This could be more thoroughly explained.

In our DMSO-differentiated protocol, we see the entire population shift toward a CD11b positive expression level, suggesting the entire population is proceeding through differentiation (e.g. see Fig. 4a for histograms of undifferentiated and differentiated wild-type HL-60 cells). Since the majority of gene knockdowns (control sgRNA and specific gene targeting sgRNA) in our differentiation screen still cluster with a \log_2 fold-change near zero (Fig. 1e, middle), we expect a similar CD11b histogram in our CRISPRi library.

We believe a bigger concern is the heterogeneous population due to cell death following differentiation (e.g. as illustrated by the change in live cell density in Fig. 3c, left panel). To minimize the addition of dead cells in our migration screens, we performed a density gradient centrifugation protocol to first isolate the live cell population.

We have added the following text in section, '*Pooled CRISPRi screens identify genes that alter cell proliferation, neutrophil differentiation, and cell migration*', where we first discuss the generation of differentiated dHL-60 cells for our differentiation and migration screens,

To derive dHL-60 cells, we induced differentiation by incubating uHL-60 cells with 1.57% DMSO for five days, which provides a near-complete differentiation of the cell population into CD11b+ neutrophil-like cells¹². Viable dHL-60 cells were isolated using density gradient centrifugation to remove dead cells and cellular debris following our differentiation protocol.

With respect to specific gene perturbations that alter cell health or differentiation, we would expect these cells to have different performance in the migration assays (e.g. SPI1 knockdown). This was part of the motivation for comparing screen hits across our various assays (e.g. Fig. 1f and Fig. 2b). Gene knockdowns that were identified across all screens would be of less interest to us.

5. Does differentiation change CRISPRi performance? Expression levels of Cas9 machinery may affect performance, which could change after differentiation. Given the screens yielded results, I think it still works, but was the CD4 test performed in differentiated cells?

Our initial construction and testing of different CRISPRi constructs was only performed in the undifferentiated cells. This is noted in the main text and captions (Fig. 1a and Supplementary Fig. 1). However, CRISPRi performance does not appear to drop after differentiation. For example, our knockdown line targeting ITGB2 showed robust protein knockdown in differentiated cells relative to

wild-type expression (see Fig. R4A above). In addition, if we look at the extent of knockdown at the transcript level for FLCN and LAMTOR1, it is quite similar in undifferentiated and differentiated cells (Fig. R4B).

While we have not looked at this across a large number of gene targets, we suspect that it is better that knockdown begins in the undifferentiated HL-60 cells. This will allow for any long-lived proteins to be lost through dilution, as the cells grow and divide. This might otherwise become a problem post-differentiation, where cells have become post-mitotic.

6. In Figure 4, CRISPRi is misspelled in the captions.

Thank you, this has been corrected.

References

1. Collins, S.R., Yang, H.W., Bongler, K.M., Guignet, E.G., Wandless, T.J., and Meyer, T. (2015). Using light to shape chemical gradients for parallel and automated analysis of chemotaxis. *Mol. Syst. Biol.* *11*, 804.
2. Bhattacharya, A., Wei, Q., Shin, J.N., Abdel Fattah, E., Bonilla, D.L., Xiang, Q., and Eissa, N.T. (2015). Autophagy Is Required for Neutrophil-Mediated Inflammation. *Cell Rep.* *12*, 1731–1739. 10.1016/j.celrep.2015.08.019.
3. Riffelmacher, T., Clarke, A., Richter, F.C., Stranks, A., Pandey, S., Danielli, S., Hublitz, P., Yu, Z., Johnson, E., Schwerd, T., et al. (2017). Autophagy-Dependent Generation of Free Fatty Acids Is Critical for Normal Neutrophil Differentiation. *Immunity* *47*, 466–480.e5. 10.1016/j.immuni.2017.08.005.
4. Wu, X., Xie, W., Xie, W., Wei, W., and Guo, J. (2022). Beyond controlling cell size: functional analyses of S6K in tumorigenesis. *Cell Death Dis.* *13*, 1–19. 10.1038/s41419-022-05081-4.
5. Murao, S., Gemmell, M.A., Callahan, M.F., Anderson, N.L., and Huberman, E. (1983). Control of macrophage cell differentiation in human promyelocytic HL-60 leukemia cells by 1,25-dihydroxyvitamin D3 and phorbol-12-myristate-13-acetate. *Cancer Res.* *43*, 4989–4996.
6. Ramirez, R.N., El-Ali, N.C., Mager, M.A., Wyman, D., Conesa, A., and Mortazavi, A. (2017). Dynamic Gene Regulatory Networks of Human Myeloid Differentiation. *Cell Syst.* *4*, 416–429.e3.
7. Breitsprecher, D., and Goode, B.L. (2013). Formins at a glance. *J. Cell Sci.* *126*, 1–7. 10.1242/jcs.107250.
8. Thompson, S.B., Sandor, A.M., Lui, V., Chung, J.W., Waldman, M.M., Long, R.A., Estin, M.L., Matsuda, J.L., Friedman, R.S., and Jacobelli, J. (2020). Formin-like 1 mediates effector T cell trafficking to inflammatory sites to enable T cell-mediated autoimmunity. *eLife* *9*, e58046. 10.7554/eLife.58046.
9. Metzner, C., Mark, C., Steinwachs, J., Lautscham, L., Stadler, F., and Fabry, B. (2015). Superstatistical analysis and modelling of heterogeneous random walks. *Nat. Commun.* *6*, 7516.
10. Garner, R., Skariah, G., Hadjitheodorou, A., Belliveau, N.M., Savinov, A., Footer, M.J., and Theriot, J.A. (2020). Neutrophil-like HL-60 cells expressing only GFP-tagged β -actin exhibit normal motility. *77*, 181–196.
11. Gupta, P., Gurudutta, G.U., Saluja, D., and Tripathi, R.P. (2009). PU.1 and partners: regulation of haematopoietic stem cell fate in normal and malignant haematopoiesis. *J. Cell. Mol. Med.* *13*, 4349–4363. 10.1111/j.1582-4934.2009.00757.x.
12. Rincón, E., Rocha-Gregg, B.L., and Collins, S.R. (2018). A map of gene expression in neutrophil-like cell lines. *BMC Genomics* *19*, 573.
13. Love, M.I., Huber, W., and Anders, S. (2014). Moderated estimation of fold change and dispersion for RNA-seq data with DESeq2. *Genome Biol.* *15*, 550. 10.1186/s13059-014-0550-8.
14. Sender, R., and Milo, R. (2021). The distribution of cellular turnover in the human body. *Nat. Med.* *27*, 45–48. 10.1038/s41591-020-01182-9.
15. Napolitano, G., Di Malta, C., and Ballabio, A. (2022). Non-canonical mTORC1 signaling at the lysosome. *Trends Cell Biol.* *32*, 920–931. 10.1016/j.tcb.2022.04.012.
16. Gollwitzer, P., Grützmaker, N., Wilhelm, S., Kümmel, D., and Demetriades, C. (2022). A Rag GTPase dimer code defines the regulation of mTORC1 by amino acids. *Nat. Cell Biol.* *24*, 1394–1406. 10.1038/s41556-022-00976-y.
17. Blanter, M., Gouwy, M., and Struyf, S. (2021). Studying Neutrophil Function in vitro: Cell Models and Environmental Factors. *J. Inflamm. Res.* *14*, 141–162. 10.2147/JIR.S284941.
18. Gironi, B., Kahveci, Z., McGill, B., Lechner, B.-D., Pagliara, S., Metz, J., Morresi, A., Palombo, F., Sassi, P., and Petrov, P.G. (2020). Effect of DMSO on the Mechanical and Structural Properties of Model and Biological Membranes. *Biophys. J.* *119*, 274–286. 10.1016/j.bpj.2020.05.037.
19. Yuan, C., Gao, J., Guo, J., Bai, L., Marshall, C., Cai, Z., Wang, L., and Xiao, M. (2014). Dimethyl Sulfoxide Damages Mitochondrial Integrity and Membrane Potential in Cultured Astrocytes. *PLoS ONE* *9*, e107447. 10.1371/journal.pone.0107447.
20. Campbell, S.L., and Wellen, K.E. (2018). Metabolic Signaling to the Nucleus in Cancer. *Mol. Cell*

71, 398–408. 10.1016/j.molcel.2018.07.015.

21. Saggese, P., Sellitto, A., Martinez, C.A., Giurato, G., Nassa, G., Rizzo, F., Tarallo, R., and Scafoglio, C. (2020). Metabolic Regulation of Epigenetic Modifications and Cell Differentiation in Cancer. *Cancers* 12, 3788. 10.3390/cancers12123788.

REVIEWERS' COMMENTS

Reviewer #1 (Remarks to the Author):

The authors did an excellent job at answering the points that were brought up. There are no further concerns.

Reviewer #2 (Remarks to the Author):

The authors have done an excellent job responding to our concerns. The paper is much more readable, and the changes to the title and abstract are great. Congrats to the team!

Reviewer #3 (Remarks to the Author):

I still have concerns about the off-target effects of sgRNAs. The evidence provided only ensures on-target efficacy. For example, the RNA-Seq shows a knockdown of the target gene, but the other differentially-expressed genes could be due to that knockdown or an off-target effect. Performing RNA-Seq on additional sgRNAs is over the top, but analyzing the sgRNAs through something like CasOFF Finder to rule out any clear confounding off-targets (e.g., targeting in the promoter/enhancer for another gene) would be beneficial.

Otherwise, the authors have addressed my concerns.

Response to Reviewer 3 for 'Whole-genome screens reveal regulators of differentiation state and context-dependent migration in human neutrophils'

Nathan M. Belliveau, Matthew J. Footer, Emel Akdogan, Aaron P. van Loon, Sean R. Collins, and Julie A. Theriot

Reviewer #3:

I still have concerns about the off-target effects of sgRNAs. The evidence provided only ensures on-target efficacy. For example, the RNA-Seq shows a knockdown of the target gene, but the other differentially-expressed genes could be due to that knockdown or an off-target effect. Performing RNA-Seq on additional sgRNAs is over the top, but analyzing the sgRNAs through something like CasOFF Finder to rule out any clear confounding off-targets (e.g., targeting in the promoter/enhancer for another gene) would be beneficial.

We appreciate the reviewer's concern over potential off-target effects. We should have also noted in our previous response that the library where these designed sgRNA originated¹ also used minimization of off-target sites as a constraint in sgRNA design. In the Methods section, 'CRISPRi pooled library construction,' we now note that '*For optimal library design, sgRNA were selected based on their position relative to annotated transcription start sites, expected on-target activity, and the presence of off-target matches.*'

We agree that it is worth being extra careful and have taken your suggestion to use CasOFF Finder to identify potential binding sites. We summarize the number of potential off-target sites in Table R1 below. Encouragingly, no binding sites were identified when 1 mismatch was allowed and only a subset of sgRNA showed binding sites when 2 mismatches were allowed. We have used the UCSC browser (GRCh38/hg38) and transcription start site annotations (via FAMTOM5²) to check whether any of these sites are near other transcription start sites.

For potential off-target sites with 2 mismatches, only the sgRNA targeting TLN1 showed some proximity to an annotated transcription start set (GGGCGAgCCGAGcAGCGGCGGGG, where lower case letters indicate the mismatch). This appears to be associated with the expression of NOS2, coding for nitric oxide synthase 2. While we do not have RNA-seq data for this cell line, this gene shows little to no detectable expression in our sgRNA control cell line or in related cell lines in other work³. Off-target binding to this site should therefore have little expected effect.

For those cell lines we do have RNA-seq data on (FLCN, LAMTOR1, and SPI1 knockdown lines), we also looked at 3 mismatches. We identify several binding sites that are near other transcription start sites (within ~200 bp) and below we summarize the relevant log₂ fold-change in expression between the target cell line and our control sgRNA cell line. Overall, the small number of genes identified show a small change in expression. In addition, several do not show a decrease in expression, which would have been expected if there were direct off-target activity by the dCas9/sgRNA.

Potential off-target knockdown for FLCN sgRNA

- MFSD4 (GCCCCGcCCtTGGGcCCGGGCGG); log₂ fold-change = 0.32
- MKRN1 (GCCCCGTCCcTGGGggCGGGCGG); log₂ fold-change = -0.39

- SLC44A2 (GCCCCcTCCGaGGGACCGaGGGG); \log_2 fold-change = -0.50
- PTH1R (GCCCCGTCCGcGGccCCGGGGGG); \log_2 fold-change not quantified due to little to no expression measured.

Potential off-target knockdown for LAMTOR1 sgRNA

- RIT1 (GCTGCTGTAGCAGCcCCgCtGGG); \log_2 fold-change = 0.20

Potential off-target knockdown for SPI1 sgRNA

- SCN9A (CtCtGGGCTCCTGTtGCTCAGGG); \log_2 fold-change = 2.7 but very low expression measured.

Note that the single off-target site with 3 mismatches for our control sgRNA (see Table R1) was also confirmed to not be located near any transcription start sites.

Overall, the results appear very encouraging and suggest that our experimental observations are predominantly due to on-target activity of the dCas9/sgRNA.

Table R1: Summary of off-target search using CasOFF Finder. The number of potential off-target binding sites were determined for each sgRNA where we made individual knockdown cell lines. Columns indicate the total number of binding sites when between 0-5 mismatches are allowed. Note that the single 0 mismatch sites correspond to the desired on-target binding site.

target	sgRNA	0 mismatches	1 mismatches	2 mismatches	3 mismatches	4 mismatches	5 mismatches
Control sgRNA	AGGGCACCCGGTTCATACGCNNG	0	0	0	1	33	320
GIT1	GGCGGCGCTTCCGCTCTAACNNG	1	0	0	0	25	225
FMNL1	GCCCCGTCCGTGGGACCGGGNG G	1	0	0	8	81	943
TSC1	GACTGTGAGGTAAACAGCTGNNG	1	0	0	11	166	>1000
ATIC	CTGGGTTCAGGGCGAGCGGGNNG	1	0	0	11	179	>1000
RICTOR	CGGGCTTACCTCGTACTCGGNNG	1	0	0	1	16	275
ITGA1	CGTGTTTAGGCTAAAGTCCANGG	1	0	0	3	54	510
APBB1IP	CCTTAGTCCCTTTCGCTCGNNG	1	0	0	4	23	337
CORO1A	ATCTTCAGCGGGCGAGTCCNNG	1	0	0	5	41	416
VPS29	CGACGGTGGTGGTACTGAGNNG	1	0	0	10	119	>1000
SNX17	TGCGGGGACTCGCTGAGCAGNNG	1	0	0	8	107	>1000
ITGB2	CGGTGTGCTGGAGTCTCGGNNG	1	0	1	8	108	>1000
CEBPE	GTAGGCGGAGAGGTCAATGGNNG	1	0	1	5	124	>1000
SPI1	CCCAGGGCTCCTGTAGCTCANGG	1	0	1	19	314	>1000
ARHGAP3 0	CAGGACACAATTTCTTGCCANGG	1	0	1	8	151	>1000

FLCN	GCCCGGGTTCAGGCTCTCAGNGG	1	0	1	14	121	>1000
TLN1	GGGCGACCCGAGAAGCGGCGNG G	1	0	1	5	73	685
LAMTOR1	GCTGCTGTAGCAGCACCCCANGG	1	0	3	24	270	>1000